# Hierarchical structure and memory mechanisms in agreement attraction

Julie Franck[1]*, Matthew Wagers[2]

**1** Department of Psychology, University of Geneva, Geneva, Switzerland, **2** Department of Linguistics, University of California, Santa Cruz, United States of America

* Julie.Franck@unige.ch

**Data Availability Statement:** Yes - all data are fully available without restriction. All data files are available from the Open Science Foundation database per DOI 10.17605/OSF.IO/J85SP. The data files are accessible via the following URL: https://doi.org/10.17605/OSF.IO/J85SP.

## Abstract

Speakers occasionally produce verbs that agree with an element that is not the subject, a so-called 'attractor'; likewise, comprehenders occasionally fail to notice agreement errors when the verb agrees with the attractor. Cross-linguistic studies converge in showing that attraction is modulated by the hierarchical position of the attractor in the sentence structure. We report two experiments exploring the link between structural position and memory representations in attraction. The method used is innovative in two respects: we used jabberwocky materials to control for semantic influences and focus on structural agreement processing, and we used a Speed-Accuracy Trade-off (SAT) design combined with a memory probe recognition task, as classically used in list memorization tasks. SAT enabled the joint measurement of retrieval speed and retrieval accuracy of subjects and attractors in sentences that typically elicit attraction errors. Experiment 1 first established that attraction arises in jabberwocky sentences, to a similar extent and showing structure-dependency effects, as in natural sentences. Experiment 2 showed a close alignment between the attraction profiles found in Experiment 1 and memory parameters. Results support a content-addressable architecture of memory representations for sentences in which nouns' accessibility depends on their syntactic position, while subjects are kept in the focus of attention.

## Introduction

### Theories of attraction

Attraction errors are characterized by the incorrect agreement of a target with an element that is not its grammatical controller (e.g., The label on the bottles *are* rusty). The agreement targets are typically verbs, pronouns or adjectives; the attracting elements nouns or pronouns; and the agreement features can be number or gender, although most of the work has concentrated on number attraction in subject-verb agreement. The phenomenon of attraction is directly observable in language production, since it gives rise to a grammatical error, and it is indeed in production that attraction was first theorized [1] and that it was for the most part explored experimentally [2].

The Marking and Morphing model of agreement production distinguishes between two causes of errors ([3], [4], and much subsequent work). The first cause lies in the conceptual

**Funding:** The research was funded by the Fonds National Suisse de la Recherche (International Short Visit) to J. Franck. http://www.snf.ch/fr/Pages/default.aspx The funders had no role in study design, data collection and analysis, decision to publish, or preparation of the manuscript.

**Competing interests:** The authors have declared that no competing interests exist.

representation of the subject's notional number. Semantic influences come about during Marking, the process during which the speaker translates the number notion from the message into a linguistic feature. Attraction may also arise at the level of Morphing, due to the influence from the attractor's feature on the highest subject node, by way of a feature percolation mechanism. In this view, each morpheme within the subject phrase is a source of number information with a particular weight, which will determine the strength of its influence on the final number value of the subject phrase. The subject head, having the biggest weight, usually wins in imposing its feature, and the risk of contamination from a modifier situated in the subject phrase depends on the depth of its embedding, which defines the length of the path it has to percolate. This accounts for the fact that, when attractors are more deeply embedded, they trigger less attraction, as shown by Bock & Cutting [5], who compared attractors situated in a relative clause attached to the subject head to those situated in a subject PP modifier [5]. It also accounts for the fact that elements situated higher in the subject phrase ('flights' in *The helicopter for the flights over the canyon are low*) generate more errors than those situated lower ('canyons' in *The helicopter for the flight over the canyons are low*) [6], [7].

Although most studies have focused on attraction from prepositional phrase modifiers situated in the subject phrase, various studies have shown that nouns that are not in preverbal position may also trigger attraction. For example, attraction was found with plural objects in the production of various structures involving movement of the object in preverbal position, like in object relatives [8], [2], [9], [10], object clefts [11] or object questions [12] In these structures, the attractor disturbs agreement even though the object is not part of the subject phrase and the subject and the verb are contiguous in the linear word string (e.g., 'patients' in *John speaks to the patients that the medicine cure*). Although the early formulation of Marking and Morphing assumed that only features from the subject phrase had the potential to influence the Morphing process [5], [3], it was later argued that any element within a structure could influence it, at a degree that depends on its structural distance from the subject [4]. However, the possibility that any element in the sentence may attract verbal agreement may also be captured by the hypothesis that attraction arises from the erroneous operation of a content-addressable mechanism responsible to retrieve the agreement controller from memory [13], [14]. This hypothesis capitalizes on a wide array of findings showing effects of similarity-based interference in the processing of sentences with long-distance dependencies, attesting to the involvement of a content-addressable, cue-based mechanism responsible for retrieving the distant element when reaching the verb [15–19]. Badecker and Kuminiak argued that such a process is also at play in agreement computation during production: indeed, in order to inflect the verb with agreement, the subject needs to be retrieved from memory, a process that is guided by cues to subjecthood (like nominative case, occupying a specifier position in the extended projection of the verb phrase, being pre-verbal). In this view, the presence, in memory, of an element bearing some similarity to the subject is the cause of an occasional erroneous retrieval of that element as the controller of agreement. As a result, attraction can potentially arise from any element in the sentence that bears some similarity with the subject. In this view, attraction with a preverbal object arises because it is available in memory when agreement needs to be computed, and it is similar to subjects in various respects: it is an NP, it bears agreement features, it occupies a similar phrase structure position (a specifier) as well as the typical sentence-initial position of subjects.

The hypothesis that cue-based retrieval is involved in the processing of agreement gained further support by studies of agreement in sentence comprehension. A number of studies using self-paced reading, eye-tracking and ERP methods have reported effects of a mismatching attractor (i.e., an attractor with a number feature that mismatches that of the subject head) taking the form of an illusion of grammaticality: in ungrammatical sentences, the presence of a

plural attractor noun mismatching the singular head but agreeing with the verb decreases the perturbation normally found in ungrammatical sentences (e.g., *The musicians who the reviewer praise so highly (. . .)*, [14], [20– 23]). Reports from the literature show that the influence of a mismatching plural attractor in grammatical sentences is generally absent, but when it is present, it also shows up in terms of faster reading times [12] or reduced comprehension errors as compared to matching attractors [24–27]. For example, Villata et al. [24] found that participants were less likely to erroneously respond 'yes' to the question *Did the dancer criticize the waiter*? when the sentence contained a mismatching attractor ('dancers' in *The dancers that the waiter strongly criticizes most of the time ordered a rum cocktail*) than when it contained a matching attractor ('dancer' in *The dancer that the waiter strongly criticizes most of the time ordered a rum cocktail*). Wagers et al. [14] argued that the existence of an effect-size asymmetry between grammatical and ungrammatical sentences was actually expected under the hypothesis of a cue-based retrieval process in comprehension, supported by the same basic memory mechanisms as Badecker & Kuminiak [13] proposed for production. According to this view, the agreeing verb in comprehension supplies retrieval cues to check for a controller NP in the parse. Such cues are expected to include information about the grammatical number of the candidate NP as well as its case or syntactic position. If the same-clause subject NP does not match the verb in number, then no single NP will fully match the retrieval cues. But the presence of a plural attractor in the parse would partially match the cues, allowing the parser to (erroneously) satisfy the agreement requirement on some proportion of trials. When a subject NP that matches the verb is present, then the correct controller of agreement will fully match the cues and any attractor will only partially match it. As a result, the effect of a mismatching number feature in grammatical strings is expected to be weaker compared to ungrammatical strings, and to manifest in terms of processing facilitation. These predictions were borne out by both behavioral evidence, as already reviewed, and computational simulations [21].

It is important to note that in sentence production, the cue-based retrieval mechanism cannot rely on agreement cues on the verb to retrieve the subject, since no such feature is present on the verb, such that other cues are assumed to be used for subject retrieval (being an NP, carrying nominative case, occupying a particular phrasal or linear position). Hence, whereas in sentence comprehension similarity-based interference can manifest in terms of penalty due to similarity in agreement features of the subject and the attractor, no such penalty can manifest in sentence production. In fact, any subject retrieval error in production studies can only show up in sentences involving a feature mismatch between the subject and the attractor, such that the effect of agreement feature similarity in production can actually simply not be studied with classical production tasks, even though much of the production literature has capitalized on that effect [28]. Nevertheless, similarity in terms of other features (semantic, syntactic, morphological) can be studied, and has been shown to negatively affect production ([29] see General discussion). Importantly, these similarity effects are left unaccounted by the Marking and Morphing model, while they find a natural explanation under the memory-based account of attraction.

Recent studies have started to cast doubt on the hypothesis that the only locus of similarity-based interference effects discussed in the sentence comprehension literature is the mechanism of memory retrieval. Indeed, memory involves not only retrieval but also encoding, and the effects reported in most of those studies can arise in either one or even both [30]. Teasing apart a retrieval locus from an encoding locus requires that similarity be manipulated on features that cannot be used as retrieval cues; if interference is found, this would provide evidence in favor of an encoding locus of the effect. Recent evidence suggests that indeed, similarity also affects encoding, since number similarity-based interference was found in acceptability judgments of verbal agreement in English past tense sentences, in which the verb carries no

number feature, while gender similarity-based interference was found in Italian, in which the verb carries no gender feature [24]. The authors argued that similarity effects arise both at encoding and at retrieval through a single mechanism of Self-Organized Sentence Processing (SOSP) responsible for building structures in memory. In SOSP, words activate treelets composed of semantic and syntactic features, which bond in virtue of their similarity. As a result, the more an attractor shares features with the subject, the more it will compete with it to occupy its position in the structure, and thus the more it will generate attraction errors (see [31] for a detailed description of the mechanism).

In sum, recent accounts of attraction have highlighted the role of memory in the processing of agreement and attraction, through mechanisms of cue-based retrieval or self-organized sentence processing, and the possibility that memory for sentences is structured hierarchically. But what do we know about the memory system that underlies sentence processing?

## Memory for sentences

There is considerable evidence that access to information in long-term memory is mediated by some form of content-addressability [32], [33], in which cues to the target memory are compiled at the retrieval site based on a subset of the relevant information available when memory is queried. The cues make contact to those memory representations in a global fashion [34]: the degree to which the cues match or are associated with target memories is evaluated simultaneously across all representations in memory, without recourse to a sequence of searches through irrelevant memories [35]. The efficacy of this process is determined by whether there are encodings that match cues, how strongly they match the cues, and how uniquely they do so [36]. Ideally, the cues would distinctly point to only the relevant, desired items.

An important source of evidence implicating content-addressability comes from studies showing that retrieval time is independent of the number of items in memory. The response-signal speed-accuracy trade-off paradigm (SAT), a procedure that investigates the full time-course of processing, enables joint measurement of retrieval speed and retrieval accuracy [37], [35]. In a probe recognition task, participants are trained to respond to a signal presented at varying time points after the onset of the recognition probe, spanning the full time course of retrieval between about 100 ms to 3000 ms. In this task, accuracy is shown to be a function of retrieval time with the following typical phases: an initial phase of chance level performance, followed by a phase of increasing accuracy, followed by an asymptotic period [32]. The asymptote provides a measure of the ultimate probability of correct retrieval, since additional time does not improve performance. Retrieval speed is measured by the intercept of the function, indicating when information first becomes available, and by the rate of rise, indicating the rate at which accuracy grows from chance to asymptote. These two parameters provide key indicators of the dynamics of retrieval, independently of the quality of memory representations.

In typical probe recognition tasks in which the SAT function is measured for judgments on each serial position in the list, asymptotic accuracy is a smoothly-varying function of position in the list [35], [38]: the asymptote increases gradually as the probed item becomes more recent in the list. On the other hand, dynamics parameters are typically not smoothly varying and instead show a simple bifurcation. For word lists, the most recent word usually enjoys a dynamics advantage over all other words [35], but the other words do not vary in terms of their retrieval speed. This has been taken as evidence for a bipartite architecture for working memory: most information remains in long term memory, while a focus of attention allows privileged access to a restricted set of representations [39], [17] [32], [40].

While research on word list memorization indicates the involvement of a cue-based retrieval mechanism [35], [41–43], this is only true if the task can be solved using only the

content of individual words in the list; for example, in a probe recognition task. When relational information is explicitly tested, i.e., when temporal or spatial order has to be retrieved, then retrieval dynamics vary with list size or item position, suggesting that the items in memory are searched in a sequence [44], [45], [35]. This raises the question of how findings from list memory experiments translate to sentences. Sentence processing often requires accessing an element situated at a distance from its dependent; for example, retrieving the subject or the object of the verb. However, sentences critically differ from lists in that they involve relations between words, organized within linearized hierarchical structures. One might therefore expect order information to play a key role in sentence processing, and therefore a mechanism of search to be involved. Nevertheless, two types of evidence also point to the involvement of a direct-access, cue-based mechanism in the processing of long-distance dependencies in sentences, rather than a search mechanism [17], [46], [42], [19], [18], [47].

The first kind of evidence comes from SAT studies of sentences, where it is found that retrieval speed is independent of the length of the dependency. In such studies, participants read sentences presented one word at a time, in Rapid Serial Visual Presentation [48]. After the final word, they make a binary acceptability judgment that depends on correctly retrieving a grammatical dependent of the word. In McElree [45], [19], the final word was a verb, which had to be paired with a displaced object. For example, compare (1) to (2):

1. It was the **scandal** that the celebrity relished/\*panicked.

2. It was the **scandal** that the model believed that the celebrity relished/\*panicked.

In both of these sentences, it is necessary to retrieve the object (bolded above) to discriminate between acceptable continuations and unacceptable ones, marked here with a star. This parallels probe recognition studies of word lists, except the nature of the probe, i.e., the verb, is itself quite different: it requires identifying an element it stands in a particular syntactic relation with. Nonetheless, McElree [45], [19] both found that the results were strikingly similar to word-list memorization studies: the distance between the probe and the target affected the asymptotic accuracy of the acceptability judgment task but processing speed was constant. The speed of retrieving a displaced object seems to be independent of the distance and the number of distractors that separate it from the verb. Similar findings have been made for the resolution of pronouns [49] and of ellipsis [50], [51]. Studies of subject-verb dependencies are also generally consistent with this picture, with one exception: when the subject and verb are linearly adjacent, they can be linked more quickly than when they are separated ([19], [40]; compare: *The editor laughed/\*ripped* vs. *The editor that the book amused laughed/\*ripped*). This adjacency advantage is akin to a focus-of-attention effect: when no other clauses intervene between subject and verb, the subject remains in the focus of attention.

The second type of evidence supporting direct, content-based access to memory during sentence processing is that retrieval accuracy is sensitive to similarity-based interference. Various studies have shown that the presence of distractors sharing some of the features of the target to-be-retrieved penalizes sentence processing. Dual-task studies in which participants are asked to memorize a list of words while reading a sentence show longer reading times and decreased comprehension accuracy when the distant target shares semantic/referential properties with words from the list [52], [53]. Various studies also showed effects of similarity between target and distractors within the sentence. Evidence from SAT and eye-tracking experiments converges to show that semantic similarity affects the processing of long-distance dependencies [54–58]. Also, the processing of long-distance dependencies in the presence of distractors that are semantically similar to the target is harder and sometimes even impossible (e.g., center-embedding), unless additional distinctive cues, like for example case markers, are

present allowing to overcome the strong semantic overlap [56]. Syntactic similarity also plays a role: a constituent intervening between the subject and the verb triggers significantly more interference when it also occupies a subject position (in a relative clause) compared to when it occupies a non-subject position [59], [60], [58], [61]. Importantly, when experimental measures allow teasing apart accessibility and the dynamics of retrieval through the SAT methods, evidence shows that semantic and syntactic similarity affect the accessibility of the element to be retrieved, but not the speed with which it is retrieved [58], [61]. This, again, supports the hypothesis that constituents are retrieved on the basis of their content, by way of a cue-based retrieval mechanism.

In sum, research on memory underlying sentence processing has revealed an important role for a cue-based retrieval mechanism in the processing of long-distance dependencies. Evidence shows that the probability of correctly accessing an element previously encountered in the sentence depends on how closely its syntactic and semantic content matches the retrieval probe (i.e., a verb that an element relates to) and on how closely other distracting elements in memory match it. Like list memorization, representations may occupy two different states: a passive state and an active state within the focus of attention. Sentential subjects stay in the focus of attention when they are adjacent to their verb, but may be shunted from the focus of attention when material intervenes. The exact conditions that underlie such shifts remain to be precisely determined [40]. Crucially, there is little evidence that a search mechanism is deployed, even when order is relevant to parsing ([19] Experiment 3; but see [21]).

## Overview of the study

The present study explores the relation between the structural conditions of attraction and memory by testing the hypothesis that elements that trigger high rates of attraction do so because they are easier to retrieve from memory than those that trigger lower rates of attraction. Until now, theoretical accounts in which memory is granted a key role in attraction rely on indirect evidence for this role; here, we explore it directly. To do so, noun phrases' availability and the dynamics of their retrieval were measured by way of a standard memory task, i.e., probe recognition, and then linked to their attraction rates measured with a classical grammaticality judgment task.

We investigated two structures previously studied with natural sentences, each containing two attractors: complex subject modifiers involving two PPs and complex object questions involving a moved object head and its PP. In these structures, the two attractors were found to give rise to different degrees of attraction in sentence production. As already discussed above, in the presence of two PPs, the hierarchically higher one ('flights' in *The helicopter for the flights over the canyon are low*) generates more interference than the lower one ('canyons' in *The helicopter for the flight over the canyons are low*, Franck et al., 2002[6]). Similarly, in complex object questions, attraction from the head of a moved complex object, situated higher in the tree ('patients' in *Quelles patientes du médecin dis-tu que l'avocat defendent*? *Which patients of the doctor do you say that the lawyer defend?*) is stronger than attraction from the same lexical element when it is in a lower position of modifier of the head ('patients' in *Le médecin de quelles patientes dis-tu que l'avocat defendent*? *The doctor of which patients do you say that the lawyer defend?*). The relevance of testing these two structures lies in a key difference with respect to the structural position of the higher attractor. In complex object questions, the hierarchically higher object head occupies a special position of c-commanding the verb [62], [63]: a node X c-commands Y iff Y is dominated by the sister node of X. The c-commanding relation has consequences on a variety of morphosyntactic and interpretive processes, including agreement (but also the binding of anaphors, the determination of quantifier

scope, etc.). In contrast, in sentences with double modifiers, the two PPs occupy positions that linearly precede the verb, without entertaining any structural relation to it.

Our aim was to determine whether the attraction potential of hierarchically higher elements (highest PP, c-commanding object head) aligns with their accessibility/speed of access from memory, which would attest a direct link between attraction and memory, a piece of evidence that has not yet been shown. To reach that goal, we designed two experiments on the processing of sentences in a semi-artificial jabberwocky language in which pseudo-nouns replaced nouns, while function words and verbs were real words of the French language. The aim of using jabberwocky was to explore the influence of structural factors on memory, and control for semantic influences on attraction, given the known influence of semantic factors like the notional plurality of the sentence [64], [65], [31]; [66], but also the semantic similarity between the subject and the attractor [67] and the semantic plausibility of the attractor as being an agent[68]. Using pseudo-nouns while preserving verbs and the grammatical skeleton of natural sentences prevents participants from building a rich semantic representation for the sentence while still allowing them to build a parse tree and computing the agreement dependency without difficulty. Moreover, given our interest in the memory for the noun phrases in the sentence, it was important to control for their lexical frequency, given its well-known influence on memory processes [69]and sentence processing more generally [70]. The first structure involves complex object questions, in which the object consists of a head and a PP modifier, as illustrated in (3). The second structure involves double PP modifiers, as illustrated in (4). The position of the mismatching plural feature was either on the hierarchically higher attractor, as in (3a) and (4a) or on the lower one, as in (3b) and (4b).

(3) a. Quels dafrans du brapou dis-tu que le bostron defend?

Which-P dafrans-P of the-S brapou-S do you say that the-S bostron-S defends-S?

b. Le brapou de quels dafrans dis-tu que le bostron defend?

The-S brapou-S of which-P dafran-P do you say that the-S bostron-S defends-S?

(4) a. Le bostron des dafrans du brapou dort.

The-S bostron-S of the-P dafrans-P of the-S brapou-S sleeps-S.

b. Le bostron du dafran des brapous dort.

The-S bostron-S of the-S dafran-S of the-P brapous-P sleeps-S.

Experiment 1 examines whether attraction arises in jabberwocky, and whether it shows sensitivity to structure as found in natural sentence production. We used a speeded grammaticality judgment task, which has been shown to consistently replicate attraction effects found in sentence production, including their sensitivity to syntactic structure ([12] Experiments 2 and 3), irrespectively of whether the sentence is grammatical or not ([12] in French, and [71], in German).

Experiment 2 tests the ease with which the noun phrases of the sentence, i.e., the subject and the two attractors, are retrieved from memory. The study uses a procedure combining a probe recognition task at the end of the sentence with the Speed-Accuracy Trade-Off design [35]. Participants saw the sentences word by word in Rapid Serial Visual Presentation mode, followed by the presentation of a pseudo-noun (the probe) that could either be the subject, the higher attractor, the lower attractor or a foil. Participants judged if the probe occurred in the sentence at each of 18 tones presented at 250 ms intervals after the onset of the last word (the verb). Response accuracy was thus measured across the full time-course of retrieval and discriminative speed-accuracy curves were estimated. This procedure is similar to that used in studies using SAT to explore list memorization [35], [41], [38], but contrasts with studies that have made use of the SAT procedure in sentence comprehension, which involve an acceptability judgment task at the sentence-final verb where a syntactic dependency is established (e.g., retrieving a displaced object or integrating a distant subject, [19], [58], see [40]for a review). In

these sentence studies, the task involves not only the memory retrieval of the distant element, but also various processes at play in judging whether the sentence is acceptable or not. Here, the use of the probe recognition task provides a measure of how accessible an element from the sentence is in memory, and of how fast it is retrieved.

One may object that the probe recognition task could be performed by a general assessment of the probe's familiarity, without the recovering of specific source information that is assumed to take place in retrieval in sentence processing. That is possible, but as familiarity is itself a way of talking about the strength of the probe's representation (i.e., its activation), the results will nonetheless be relevant. If our participants could perform this task without recovering source information, then we may lose information about the dynamics of retrieval from different syntactic positions. However, other SAT studies testing the relative combination of familiarity and source information have found that probe recognition tasks are usually accomplished by a combination of both [72], [73]. Moreover, we don't think our participants could respond accurately without recovering source information: in our study design the same probe occurred on multiple trials, either as the positive or negative probe–so from trial to trial familiar probes either have to be rejected or accepted, depending on whether or not they were in the most recently processed sentence.

The use of the SAT paradigm was expected to inform us about two aspects of items in memory: the overall strength or accessibility of that item, which we link to the construct of activation, and the speed with which that item can enter the processing stream, which has been linked to the concept of focus of attention [19]. Because judgments about a probe are collected across an extended time-span, it is possible to estimate the function that relates task performance to time elapsed. As a consequence, it is possible to determine the maximum attainable accuracy for a task and the rate at which information accrues. A good model of typical SAT data is given by the shifted exponential function below, in which performance is measured by d-prime (d′):

$$'(t) = \begin{cases} \lambda(1 - e^{-\beta(t-\delta)}), & t > \delta \\ 0, & elsewhere \end{cases}$$

This equation can be understood by identifying three periods in task performance:

- A period of chance performance, indexed by the x-intercept parameter δ (ms);

- A period of increasing accuracy, indexed by the rate parameter β (ms$^{-1}$);

- A period in which performance approaches its limit, indexed by the asymptote parameter $\lambda$ (in units of d').

The first two parameters (δ, β) serve as a joint measurement of memory access dynamics, and can be used to ask whether certain items in memory take longer on average to access [38], [19]. The inverse of the rate parameter β is the function's time constant, and when $t = \frac{1}{\beta} + \delta$ (ms), then d′ has achieved approximately 63% of asymptotic accuracy (or, exactly $100 \cdot (1-e^{-1})$%). The asymptote parameter λ relates to the availability or strength of an item in memory, which controls the maximum attainable task performance.

Our predictions can be summarized as follows. If the structural component of attraction is syntactic in nature and independent of the semantic content of the words and their relationships, we expect Experiment 1 to replicate the findings from natural language in our jabberwocky materials:

**Prediction 1.** Generally more errors and/or slower response times in the presence of a mismatching plural attractor;

**Prediction 2.** Stronger attraction from a plural attractor situated high and/or c-commanding the verb than from a plural attractor situated low and intervening by precedence on the agreement relation.

If our hypothesis is correct that memory access for elements from the sentence operates on hierarchical representations and that agreement/attraction is tightly linked to the properties of items in memory, then we expect the following in Experiment 2:

**Prediction 3.** Subject heads, with which agreement is usually computed correctly, should be more accessible (higher λ) and/or retrieved earlier/faster (lower δ, higher β) than the two attractors, regardless of their position in the linear word string;

**Prediction 4.** Attractors that trigger high attraction, expected to be those in a hierarchically higher position and/or occupying a c-commanding position with regard to the verb, should be more accessible (higher λ) and/or retrieved earlier/faster (lower δ, higher β) than those that trigger weaker attraction.

Predictions 3 and 4 are agnostic about whether effects of prominence in memory will be reflected in accessibility or retrieval speed. We adopt this agnosticism for two reasons. Firstly, the bi-partite model of working memory advocated by McElree [32], if taken at face value, suggests that changes in the activation of particular encodings should only affect the ultimate likelihood of retrieval, and not retrieval speed. The retrieval speed, in this view, is affected primarily by whether or not the required information is in the focus of attention, or not. However, we understand so little at present about how linguistic information flows through the focus of attention, that *a priori* it is conceivable that some hierarchical effects in memory could be accounted for by how the focus of attention is managed. Secondly, ACT-R, which has been adapted to explain sentence processing [17], explicitly links activation with latency to retrieve a chunk, with higher activations mapped to shorter latencies. This model, if taken at face value, suggests that changes in the activation of particular encodings should also affect retrieval speed. How that linking hypothesis translates into SAT data is complicated, because there are more parameters than just the retrieval latency equation in a full ACT-R model. Depending on time-to-encode and response strategies, differences in activation could nonetheless be manifested in asymptotic differences only, with very small effects on dynamics [74].

## Experiment 1: Attraction in jabberwocky

### Method

**Participants.** Forty-six participants took part in the experiment. They were all native French speakers aged between 20 and 40 with no reported hearing or language impairment. They received course credit for their participation. The study was approved by the Ethics committee of the University of Geneva, and participants signed written consent forms.

**Materials.** Experimental materials consisted of jabberwocky sentences containing pseudo-nouns, real grammatical words and real verbs. Pseudo-nouns respected phonotactic constraints of the French language, and their resemblance to existing words was minimized. A total of 192 items varying along 4 dimensions were created: Structure, characterizing the nature of the attracting element (Object vs. Modifier); Height, characterizing the position of the attractor noun varying in number (Low vs. High, with the additional specificity of object heads being not only high but also c-commanding the verb); Match, characterizing the number match between the head and attractor nouns (Match vs. Mismatch); and Grammaticality (Grammatical vs. Ungrammatical). Whereas transitive verbs were used in the Object condition, intransitive verbs were used in the Modifier condition. In the Object condition, we controlled for the position of the *wh*-element such that it always appeared in the NP containing the plural feature (whether high or low). All subject heads were singular. Structure was

manipulated between-items while Height, Match and Grammaticality were manipulated within-items. Examples of items in the different experimental conditions are provided in Table 1.

Items were distributed across 4 lists and each list contained two versions of an item. There were 48 experimental items in total, combined with 48 fillers of the same type as the experimental ones, whose subject heads were all plural and which varied in Structure, Height, Match and Grammaticality (all between-items).

**Procedure.** Materials were presented on a computer screen using the E-Prime software. Sentences were split in windows corresponding to phrases (grammatical words were presented together with the content word they were linked to). Windows were presented for a fixed period of 500 ms, except at the verb, i.e., the final word of the sentence. These rather long presentation windows were chosen given the unfamiliarity of the jabberwocky words. Grammaticality judgment times (here called RTs) were measured at the verb onset. Participants were asked to judge the grammaticality of the sentences as quickly as possible and press on the corresponding response button. The experimenter illustrated the task by presenting 6 example sentences and explaining why they could be considered grammatical or ungrammatical. The experiment started with 6 practice trials and a pause was introduced in the middle of the experiment.

While participants had no difficulty understanding the task, they gave no response within 5000 ms of the verb onset in 12% of trials and no response was recorded. The no-response trials were uniformly distributed across Match, Height and Grammaticality factors; but there were significantly more no-response trials for Object structures than for Modifier structures ($\chi^2(1) = 28$, $p < .001$). These time-out trials were moved from further analysis. One participant was removed for very low accuracy across the entire experiment (69%, which was greater than 2 standard deviations from the mean accuracy on the logit scale) and another was removed because their session was truncated (producing fewer than 10 trials).

**Data analysis.** Regression models of dependent variables (Accuracy, RTs) were carried out on the experimental factors (Structure, Height, Grammaticality, and Match) and their

**Table 1. Examples of items in the different experimental conditions formed by the crossing of structure, height, number match and grammaticality in Experiment 1.**

| Structure | Height | Match | Example |
|---|---|---|---|
| Modifier | High | Match | Le bostron du dafran du brapou dort/*dorment |
| | | | The bostron of-the-sg dafran of-the brapou sleeps/*sleep |
| | | Mismatch | Le bostron des dafrans du brapou dort/*dorment |
| | | | The bostron of-the-pl dafrans of-the brapou sleeps/*sleep |
| | Low | Match | Le bostron du brapou du dafran dort/*dorment |
| | | | The bostron of-the brapou of-the-sg dafran sleeps/*sleep |
| | | Mismatch | Le bostron du brapou des dafrans dort/*dorment |
| | | | The bostron of-the brapou of-the-pl dafrans sleeps/*sleep |
| Object | High | Match | Quel drafran du brapou dis-tu que le bostron defend/*défendent? |
| | | | Which dafran-sg of-the brapou do you say that the bostron defends/*defend? |
| | | Mismatch | Quels drafrans du brapou dis-tu que le bostron defend/*défendent? |
| | | | Which dafrans-pl of-the brapou do you say that the bostron defends/*defend? |
| | Low | Match | Le brapou de quel dafran dis-tu que le bostron defend/*défendent? |
| | | | The brapou of-which-sg dafran-sg do you say that the bostron defends/*defend? |
| | | Mismatch | Le brapou de quels dafrans dis-tu que le bostron defend/*défendent? |
| | | | The brapou of-which-pl dafrans-pl do you say that the bostron defends/*defend? |

interactions. Single factors were sum-coded to (+1/2, -1/2) and the following levels had positive valence: Structure:*Modifier*, Height:*High*, Grammaticality:*Ungrammatical*, and Match: *Mismatch*. For RTs, we used mixed-effects models with maximal random effects where possible and, where the model failed to converge, we attempted to include critical slopes [75]. Models were estimated using the *lme4* package in R [76]; *p*-values are reported from the *lmerTest* package [77] using the Satterthwaite approximation. For Accuracy, we used Firth's penalized likelihood method [78], implemented in the R package *logistf* [79] and confidence intervals reported based on penalized profile likelihood. This analysis is more appropriate than mixed-effects logistic regression in this application, because we observe quasi-complete separation due to (nearly) perfect performance in some conditions, such as High Modifier and High Object Match conditions. Moreover, the median number of errors per participants was 3, and many participants made only one error (n = 10) or no errors (n = 5).

## Results

Attraction was present in both Object and Modifier jabberwocky sentences, as shown by the accuracy in the grammaticality judgment task, given in Table 2. The overall error rate was 7%, ranging from less than 1% for Grammatical Object High Match conditions, to 16–18% in Object and Modifier Mismatch conditions.

Table 3 gives the results of the logistic regression, which reveals that there were significant effects of Match, Structure, Match × Height, Match × Height, and, critically, Match × Height × Structure. To resolve the source of this interaction, we conducted separate regressions on Object and Modifier data. For Object conditions, there was an effect of Match ($\beta_0$ = 3.22; $\beta_{Match}$ = -1.28, 95% C.I. [-2.2, -0.51], p = .001), an interaction of Match and Height ($\beta_{Match \times Height}$ = -2.36, [-4.1, -0.83], p = .002), and no other effects achieved significance. For Modifier conditions, there was an effect of Match ($\beta_0$ = 2.42; $\beta_{Match}$ = -1.28, [-1.85, -0.78], p < .001), of Height ($\beta_{Height}$ = 0.52, [.01, 1.08], p = 0.047), and a marginal effect of Grammaticality ($\beta_{Grammaticality}$ = -0.46, [-1.02, .06], p = 0.082). Crucially there was no interaction of Match and Height ($\beta_{Match \times Height}$ = -0.36, [-1.48, 0.76], p = .498) in Modifier conditions. Only in Object conditions was attraction, as measured by a Match effect on accuracy, influenced by the height of the attractor.

RTs paint a more complex picture, and they are reported in Fig 1. We only analyzed correct responses, because there were relatively few error trials. Although these are trials in which participants did not make an error in their overt response, their RTs did indicate that a Mismatching attractor influenced the grammaticality judgment time. Table 4 gives the results of the

**Table 2. Percentage of correct responses in the grammaticality judgment task of experiment 1.**

| Structure | Height | Match | Grammaticality | |
|---|---|---|---|---|
| | | | Grammatical | Ungrammatical |
| Modifier | High | Match | 98 (1) | 96 (2) |
| | | Mismatch | 90 (3) | 84 (3) |
| | Low | Match | 95 (2) | 93 (3) |
| | | Mismatch | 85 (4) | 82 (4) |
| Object | High | Match | 99 (1) | 99 (1) |
| | | Mismatch | 90 (4) | 84 (4) |
| | Low | Match | 97 (2) | 97 (1) |
| | | Mismatch | 98 (1) | 96 (2) |

Standard error of the mean is reported in parentheses.

**Table 3. Logistic regression on grammaticality judgment accuracy in Experiment 1.**

| | Coef | SE | 95% C.I. | | |
| --- | --- | --- | --- | --- | --- |
| | | | Lower | Upper | |
| (Intercept) | 2.82 | 0.12 | 2.6 | 3.08 | *** |
| Match | -1.28 | 0.25 | -1.79 | -0.82 | *** |
| Structure | -0.80 | 0.25 | -1.31 | -0.33 | *** |
| Height | 0.14 | 0.25 | -0.33 | 0.65 | |
| Grammaticality | -0.37 | 0.25 | -0.87 | 0.11 | |
| Match:Structure | 0.0 | 0.49 | -0.95 | 1.01 | |
| Match:Height | -1.36 | 0.49 | -2.38 | -0.43 | *** |
| Match:Grammaticality | -0.25 | 0.49 | -1.23 | 0.73 | |
| Structure:Height | 0.76 | 0.49 | -0.24 | 1.72 | |
| Structure:Grammaticality | -0.17 | 0.49 | -1.15 | 0.81 | |
| Height:Grammaticality | -0.12 | 0.49 | -1.1 | 0.86 | |
| Match:Structure:Height | 2.0 | 0.99 | 0.11 | 4.02 | * |
| Match:Structure:Grammaticality | 0.7 | 0.99 | -1.24 | 2.68 | |
| Match:Height:Grammaticality | 0.37 | 0.99 | -1.57 | 2.35 | |
| Structure:Height:Grammaticality | -0.61 | 0.99 | -2.59 | 1.33 | |
| Match:Structure:Height:Grammaticality | -0.12 | 1.97 | -4.04 | 3.80 | |

Coefficients and profile likelihood confidence intervals are reported. Additionally, asterisks indicate significance at conventional levels (

*** $p < .001$

* p < .05).

linear regression, which reveals that there were significant effects of Match, Grammaticality, Structure × Height, and Match × Structure × Height.

To make sense of the two significant interactions with Structure, we carried out separate regressions on just the Modifier conditions and just the Object conditions. In the Modifier conditions regression model, there were significant effects of Match ($\beta_0$ = 500 ms; $\beta_{Match}$ = 199 ms, 95% C.I. [129 ms, 274 ms]; t = 5.8, p < .001), Grammaticality ($\beta_{Grammaticality}$ = 109 ms, [44 ms, 181 ms]; t = 3.2, p = .002), and Height ($\beta_{Height}$ = -72 ms, [-136 ms, -5 ms]; t = -2.1, p = .037). Here, the interaction of Match with Height was significant ($\beta_{Match \times Height}$ = -167 ms, [-287 ms, -30 ms]; t = -2.4, p = .015); its sign reflects the fact that Low, Modifier conditions showed a greater Match effect than High, Modifier conditions. There was also a marginally significant interaction with Grammaticality and Height ($\beta_{Gramm \times Height}$ = -119 ms, [-237 ms, 22, ms]; t = -1.7, p = .083). By contrast, in the Object conditions regression model, the only significant effect was Match ($\beta_0$ = 512 ms; $\beta_{Match}$ = 175 ms, [83 ms, 269 ms]; t = 4.0, p < .001). Thus there was no influence of Height on Match in judgment times for Object conditions, like there was for Modifier conditions.

## Discussion

There are four key findings from Experiment 1. First, attraction arises in jabberwocky sentences, in which semantics plays virtually no role. Participants made more errors in judging the grammaticality of sentences containing a plural attractor that mismatched the number of the head noun, than in sentences containing two singular nouns. Accuracy was similar to that found in natural languages–above 80%, and nearly perfect in many conditions–suggesting that participants had no difficulty in judging the grammaticality of jabberwocky sentences. This finding supports the view that attraction arises independently of semantic factors, even if such factors can also contribute to attraction [67], [68], [80].

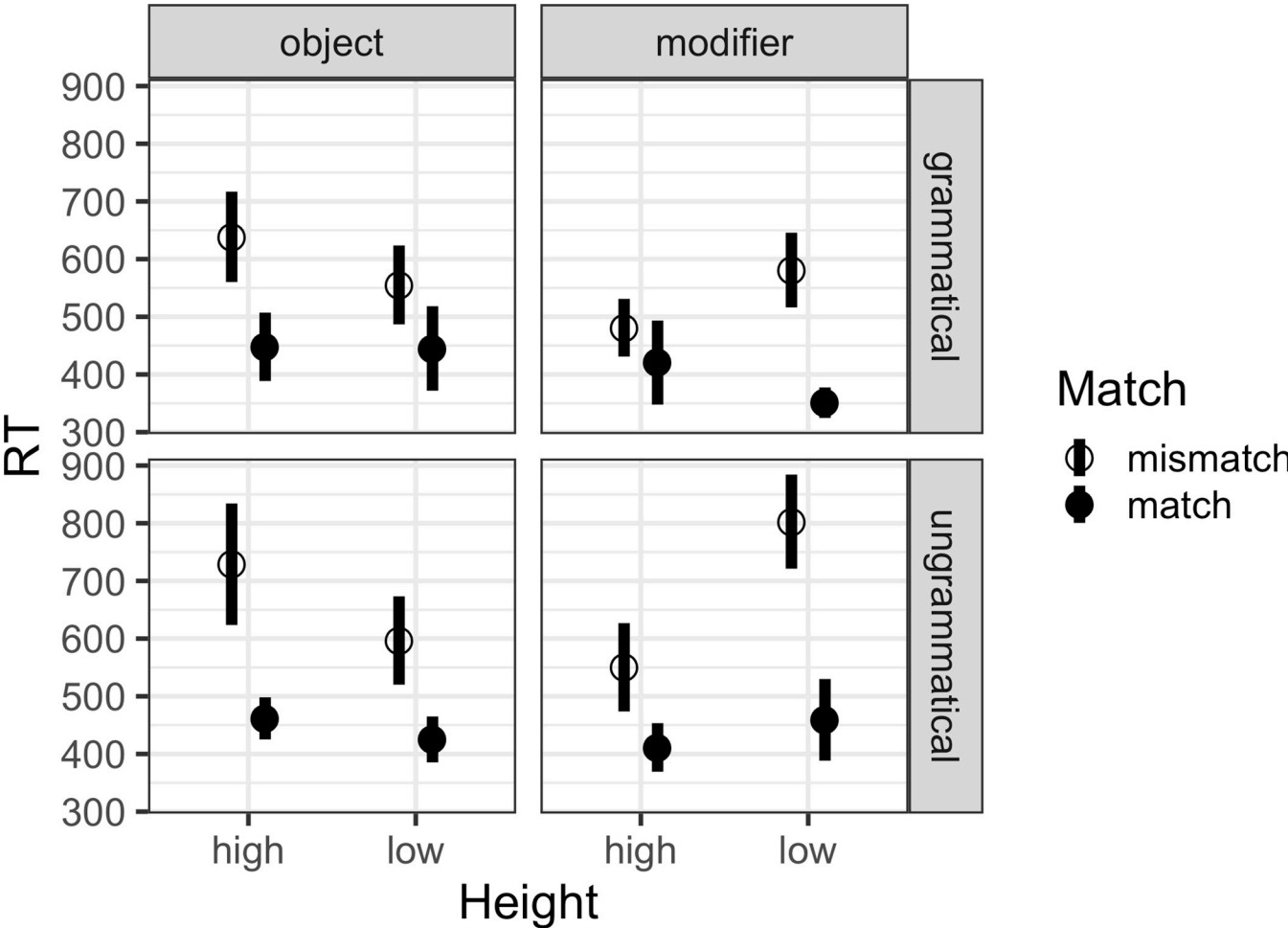

**Fig 1. Mean judgment RTs in Experiment 1.** Error bars report standard error over items.

Second, attraction was stronger when the mismatching plural noun was the head of a moved complex object, compared to when it was embedded in that constituent. Remember that in both cases the attractor was the question word or *wh*-phrase, and thus the information-structural focus of the sentence; hence, the two conditions are comparable in that regard. In High conditions, the mismatching plural phrase c-commands the verb, whereas in Low conditions, it does not; it only precedes it. This finding replicates natural language data obtained with the same structures [12], as well as other reports showing that c-commanding elements have a stronger attraction potential than preceding ones [11], [9], [81] and suggests that the hierarchical effects on attraction are a product of syntactic structure.

Thirdly, in the double modifier condition, both attractors generated similar rates of attraction. This finding contrasts with natural language data showing stronger attraction with the highest attractor [6], [7]. Given the unexpected nature of this null result, which was not predicted by our theoretical framework, we first wanted determine whether it is paralleled in Experiment 2 before discussing possible explanations.

Finally, attraction was found for both grammatical and ungrammatical sentences, and manifested as penalty due to number mismatch. This finding aligns with other studies that used a grammaticality judgment procedure in German [71] and in French [12]. However, it contrasts

**Table 4. Mixed-effects linear regression on judgment times in Experiment 1.**

| | Coef | SE | 95% C.I. | | |
| | | | Lower | Upper | |
|---|---:|---:|---:|---:|---|
| (Intercept) | 504 | 46 | 417 | 599 | *** |
| Match | 184 | 28 | 127 | 230 | *** |
| Structure | -11 | 33 | -75 | 51 | |
| Height | -14 | 28 | -65 | 44 | |
| Grammaticality | 68 | 28 | 14 | 126 | * |
| Match:Structure | 24 | 56 | -91 | 122 | |
| Match:Height | -59 | 56 | -174 | 49 | |
| Match:Grammaticality | 66 | 56 | -47 | 180 | |
| Structure:Height | -123 | 56 | -233 | -14 | * |
| Structure:Grammaticality | 87 | 56 | -25 | 184 | |
| Height:Grammaticality | -48 | 56 | -167 | 65 | |
| Match:Structure:Height | -229 | 111 | -452 | -6 | * |
| Match:Structure:Grammaticality | 39 | 111 | -180 | 245 | |
| Match:Height:Grammaticality | -21 | 111 | -229 | 186 | |
| Structure:Height:Grammaticality | -121 | 111 | -328 | 102 | |
| Match:Structure:Height:Grammaticality | -28 | 222 | -497 | 405 | |

Coefficients and bootstrap confidence intervals are reported. Additionally, asterisks indicate significance at conventional levels (*** $p < .001$, * $p < .05$).

with findings from comprehension studies based on reading time tasks in which the presence of a mismatching feature was found to facilitate the reading of ungrammatical sentences ([21], [22], [14] see [82] for a meta-analysis) as well as grammatical ones [12], [24],[25]. In line with Franck et al. [12], we suggest that the difference comes from how the two tasks may differentially encourage use of distinct cues or distinct mechanisms. In reading, the verb is presented with its agreement morphology, such that an agreement feature is salient for use in retrieving the subject to build the sentence's structure. This process is facilitated if the attractor carries a different feature, due to the lower feature overlap between the verb and the attractor. In grammaticality judgment, although the verb carries an agreement feature, this feature is erroneous in a certain proportion of the trials and the task requires that the feature be checked. Therefore, participants may attempt to check the feature by covertly producing the verb phrase, based on the subject (akin to a preamble completion task, [2]). Crucially this would also affect grammatical sentences. And as a consequence, misidentification of the attractor as the agreement controller can only lead to a detectable error when the attractor mismatches the controller, exactly like what is found in sentence production.

These four key findings in Experiment 1 all rely on the accuracy measure. Response times showed a less clear distribution: they show sensitivity to feature mismatch (with slower RTs in mismatch than match conditions), which is independent of the attractor's position in the object structure, but stronger for low than high modifiers in the modifier structure. That is, response times differ from the accuracy measure in how height affects them: for object conditions, although high, mismatching attractors, which showed significant more errors in accuracy, did also lead to numerically longer judgment times, that difference was not significant. This is not necessarily surprising, given how variable long reaction times are. For modifier conditions, low, mismatching attractors led to significantly longer judgment times, even though accuracy was not lower. Ultimately, we focus our discussion on the judgment accuracy

because we believe it is more directly interpretable. As the results of Experiment 2 will show, asymptotic accuracy in jabberwocky probe recognition tracks accuracy in jabberwocky grammaticality judgment. But why should participants' RTs be longer for low, mismatching attractors in modifier conditions if they don't lead to less accurate responses? We don't have a clear understanding of this observation yet, but we conjecture that a string-local clash between the low, mismatching modifier attractor and the main verb could be responsible ([14], for a similar low-level plural effect). In any case, it is interesting to note that recent studies which collected both accuracy and RTs also found that whereas clear effects systematically emerge in accuracy, they either fail to emerge or are reduced in RTs [24], [25].

Experiment 1 shows that attraction arises in a jabberwocky language, providing us with a tool to explore attraction with minimal influences from the semantics. It also shows that attraction with jabberwocky sentences is sensitive to the key distinction between precedence and c-command intervention, since the c-commanding attractor in object questions generated more attraction than the merely preceding one. The next step in exploring the links between attraction and memory is now to determine whether the strength of attraction found in Experiment 1 is a function of the ease with which the attractor is retrieved from memory, which is the aim of Experiment 2.

## Experiment 2: Linking attraction to memory

### Methods

**Participants.** Twenty-five participants took part in the experiment. They were all native French speakers, with ages ranging between 20 and 40, with no reported hearing or language impairment. They received course credits for their participation. The study was approved by the Ethics committee of the University of Geneva, and participants signed written consent forms.

**Materials.** A total of 432 experimental items were created, organized into 36 item sets defined by the crossing of three variables: Structure (Object vs. Modifier), Probe (Subject vs. High attractor vs. Low attractor), and Probe status (Target vs. Distractor). Each item therefore appeared in 12 conditions. The sentences contained the same pseudo-nouns as Experiment 1. All sentences were grammatical, and they always contained one plural feature (there was no match condition). The position of the plural feature was counterbalanced on the three NPs (Subject, Attractor high, Attractor low). Distractor words used for the probe recognition task were taken from the list of pseudo-nouns used for building the sentences, such that interference could occur across the board from previously seen sentences. The selection of distractor words associated to each item was randomized. Table 5 illustrates the distribution of probe

**Table 5. Illustration of the distribution of probe words in the 6 experimental conditions involving an item with the modifier structure and the plural feature situated on the subject in Experiment 2.** The same distribution applied to the corresponding 6 versions of that item in the Object condition. Probe words were either present in the sentence (Target) or not (Distractor), and if present in the sentence, it could be in 3 different positions: Subject, High attractor or Low attractor.

| Probe word position | Probe word status | Sentence | Probe word |
|---|---|---|---|
| High | Target | Les bostrons du dafran du brapou dorment | Dafran |
| | Distractor | The bostrons of the dafran of the brapou sleep | Lamolle |
| Low | Target | Les brapous du bostron du dafran dorment | Dafran |
| | Distractor | The brapous of the bostron of the dafran sleep | Rupanne |
| Subject | Target | Les dafrans du brapou du bostron dorment | Dafrans |
| | Distractor | The dafrans of the brapou of the bostron sleep | Mitelles |

words in the six experimental conditions involving an item with the Modifier structure and a plural subject.

A total of 216 additional filler items were created, also spread in 18 sets of 12 variants of each item, following the same design as experimental items. Filler items were all ungrammatical with respect to agreement. They represent 1/3 of the total items. The 648 items (432 + 216) were distributed in 6 lists of 108 items each.

**Procedure.** The multiple-response SAT procedure, as described below, was used to estimate accuracy as a function of time [83], [50]. Trials began with a 1-second fixation cross in the center of the display. Sentences were visually presented word-by-word. Word stimulus onset asynchrony varied by word length according to the formula: SOA = ArgMax(190 ms + 25 ms/char, 400ms). Inter-stimulus interval was constant at 100 ms. Fifty ms after verb offset, participants were presented with a probe word that was either one of the three nouns of the sentence or a distractor word not in the sentence. Fifty ms after the probe appeared, a series of 18 tones was presented. Each 1000 Hz tone was 50 ms in duration and there was a lag of 350 ms between the offset of a tone and the onset of the following tone. Participants were trained to press the button 'yes' if the probe was in the sentence, or 'no' if it was not. Starting from the first tone, they were trained to press the two response buttons simultaneously until they had decided whether the probe was in the sentence. Right after the probe recognition task, participants were required to perform a grammaticality judgment task, again by button pressing. This task was added to ensure that participants were set in conditions that were maximally similar to those of Experiment 1, and ensure that they would fully parse the sentence rather than develop a strategy of simply memorizing the pseudo-nouns, which was sufficient to perform probe recognition. Results to that task were not analyzed.

Participants came to the lab four times. The first session was dedicated to training. They were familiarized with jabberwocky sentences and the possible occurrence of agreement errors. They were then presented with the instructions, and progressively trained to respond contingently to the tones. The participants received feedback if they took longer than 200 ms to begin responding, if the first two responses were not simultaneously executed, or if they gave fewer than 16 total responses within 6000 ms. The next three sessions were dedicated to running the experiment proper. Two lists were presented per session, separated by a break. A session lasted about 20 minutes.

**Analysis.** For each participant, we calculated accuracy at each response tone. A d′ score was calculated by first transforming accuracy scores by the inverse normal distribution function. The resulting Z-score of the false alarm rate, i.e., percent incorrect for Distractors, was subtracted from the Z-score of the hit rate, i.e., percent correct for Targets [84]. Lag-latency was calculated by adding the average response time at each response tone to tone latency. The resulting <d′, lag-latency> series was fit by a saturating, shifted exponential function:

$$'(t) = \begin{cases} \lambda(1 - e^{-\beta(t-\delta)}), & t > \delta \\ 0, & elsewhere \end{cases}$$

This function is described by three parameters: an asymptote, $\lambda$; a rate, $\beta$; and an intercept, $\delta$. The $\lambda$ parameter describes maximum achieved performance. The speed of processing is jointly captured by the $\beta$ and $\delta$ parameters. The value of $\delta$ is the amount the curve is shifted from the ordinate axis, reflecting the moment when discriminative information is first available. Following the convention in this literature, we refer to $\delta$ as the intercept parameter, although note that it refers specifically to the x-intercept. The value of $(1/\beta + \delta)$ is the time at which accuracy reaches a common proportion of asymptotic accuracy, namely $(1-e^{-1})$, approximately 63%.

We fit a fully-saturated model to each participants' data by estimating a separate $<\lambda, \beta, \delta>$ triplet for each of the six Structure × Probe Word Target-Distractor condition-pairs. We estimated the parameters using an iterative hill-climbing algorithm [85], based on STEPIT [86], which minimizes the squared deviations of predicted values from observed data. We then use mixed-effects linear regression to test whether either asymptotic performance ($\lambda$) or rate of information accrual ($1/\beta + \delta$) varied by either Structure (Object, Modifier) or Probe Word (Subject, High, Low). While it is conceivable to separately analyze $\beta$ and $\delta$ parameters, practically they trade off during the estimation process (due to the characteristically rapid rise of the SAT function in many tasks). Therefore, the sum ($1/\beta + \delta$) is more appropriate to analyze in a fully-saturated model. We used sum contrasts for the Structure factor (+1/2 for Object, -1/2 for Modifier). For the Probe Word contrasts, we used Helmert contrasts: the first coefficient, Subjecthood, compared Subject vs. non-Subject probes, i.e. the average of High and Low conditions parameters (coded as +2/3 vs. -1/3 respectively); and the second, Height, directly compared High vs. Low non-Subject probes (coded as +1/2 vs. -1/2). For purposes of inference, we analyzed over individual participant parameters, not average data. For a convenient visualization, we additionally computed an average d-prime series over all participant data and then fit a fully-saturated model to that series. There was convergence between the two analyses.

## Results

In brief, we found that both asymptotic performance and speed were lower for non-Subjects than Subjects. Among the non-Subject attractor positions, we found a Height difference in asymptotic performance only in Object conditions. In Modifier conditions, both attractor positions were equally available in memory. There were no speed of processing differences among probes in the non-Subject positions. Fig 2 visualizes participants' parameter estimates and Table 6 summarizes them. Fig 3 visualizes the average SAT curve.

Turning first to asymptotic performance, we found that participants were highly accurate at the task and discriminated between Targets and Distractors with an average d' of 3.0. There was an overall advantage for Subject probes ($\beta_{ProbeWdSubject}$: 0.26, 95% C.I. [0.13, 0.38]; t = 4.0, p < .001), which received the highest asymptotic d' scores. There was also an overall advantage for Modifier structures compared to Object structures ($\beta_{Structure}$: -0.21, [-0.33, -0.08]; t = -3.2, p < .005). There was a significant interaction between Structure and Subjecthood ($\beta_{Strctr \times PrbWdSubj}$: 0.30, [0.15, 0.44], t = 4.0, p < .001) and a significant interaction between Structure and Height ($\beta_{Strctr \times PrbWdHeight}$: 0.28, [0.10, 0.46]; t = 3.3, p < .005). To understand the source of these interactions, we created separate models on Object and Modifier conditions. In the Object condition, there was both an advantage for Subject probes ($\beta_0$: 2.94; $\beta_{PrbWdSubject}$: 0.41, [0.25, 0.56]; t = 5.0, p < .001) and an effect of Height, with High probes attaining greater sensitivity than Low probes ($\beta_{PrbWdHeight}$: 0.24, [0.07, 0.40]; t = 2.5, p = .015). In Modifier conditions, there was a marginal advantage for Subject probes ($\beta_0$: 3.15; $\beta_{PrbWdSubject}$: 0.11, [-0.01, 0.23]; t = 1.9, p = .066), and no effect of Height ($\beta_{PrbWdHeight}$: -0.05, [-0.16, 0.09]; t = -0.70, p = .490).

We then analyzed the combined dynamics of processing. This measure sums the time constant of the fitted curve ($1/\beta$) to its intercept ($\delta$). The resulting value, in milliseconds, has a direct interpretation: it is the time since the onset of the probe at which approximately 63% of asymptotic accuracy is achieved. This value, on average, was 1433 ms. Among our experimental factors, only Probe word type affected the combined dynamics: Subject probes led to an advantage of 108 ms ($\beta_0$: 1433; $\beta_{PrbWdSubject}$: -108 ms, 95% C.I. [173 ms, 37 ms]; t = 3.1; p = .005). No other effects were significant.

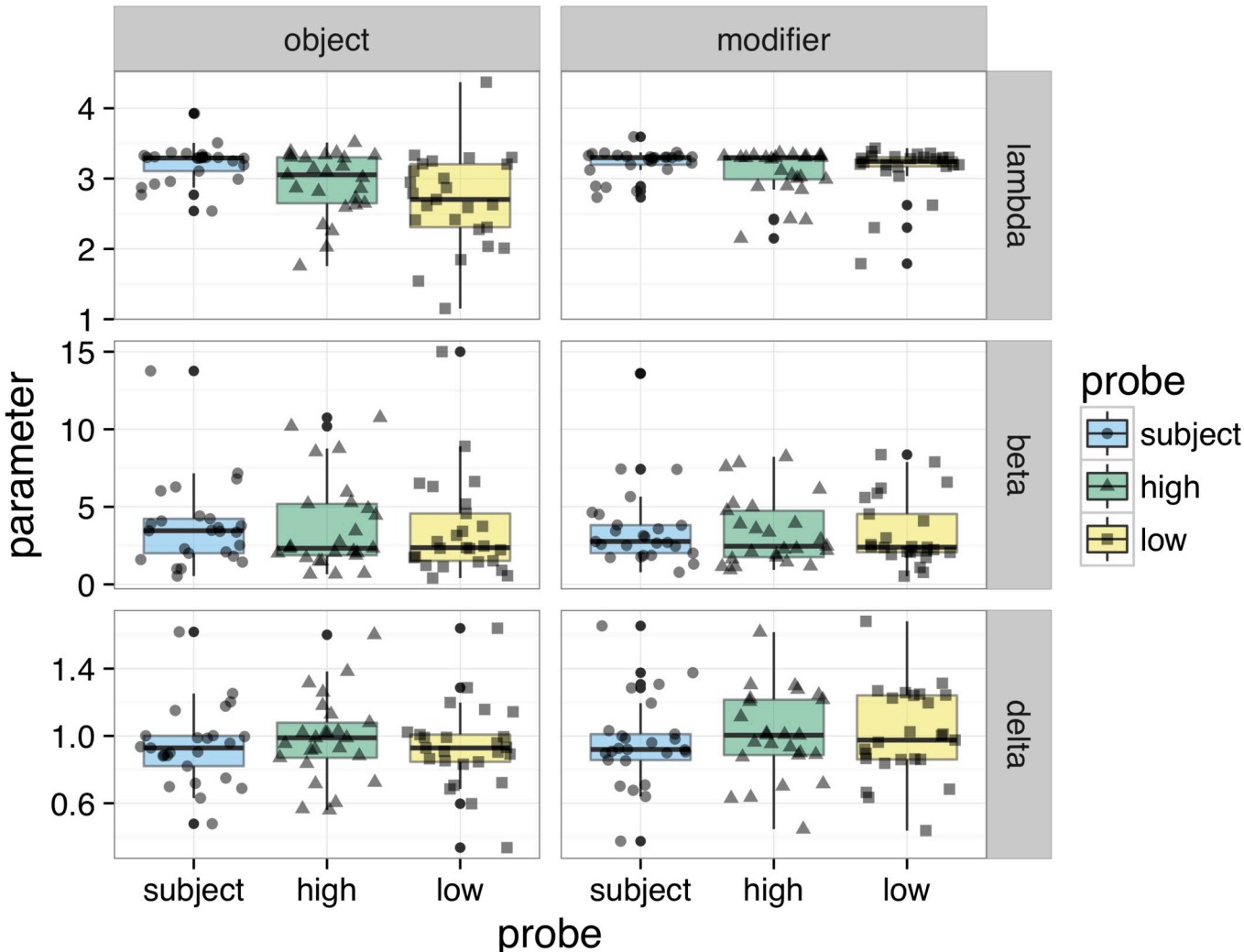

**Fig 2. Experiment 2 participant SAT parameters.** Box and scatter plots of individual parameter estimates for the asymptote, λ (d′), rate, β (ms⁻¹), and intercept, δ (ms).

Given that the major difference among conditions was in the ultimate attained accuracy, we conducted an analysis of empirical asymptotic d′ to guard against the possibility of a fitting artefact. In this analysis, we took the d′ series calculated for each condition and averaged the last four responses. We found empirical asymptotic d′ matched fit d′ quite closely. In a regression of empirical d′ on the same experimental factors, we found a mean d′ of 3.0; an overall

**Table 6. Summary of participant parameter estimates in Experiment 2.**

| Parameter | Structure | Subject | High | Low |
|---|---|---|---|---|
| Asymptote λ (d′) | Object | 3.2 (.05) | 2.9 (.09) | 2.7 (.14) |
| | Modifier | 3.2 (.04) | 3.1 (.07) | 3.1 (.07) |
| Dynamics 1/β + δ (ms) | Object | 1373 (107) | 1467 (104) | 1495 (128) |
| | Modifier | 1349 (88) | 1437 (80) | 1477 (109) |

Mean values are reported in each cell with standard errors in parentheses.

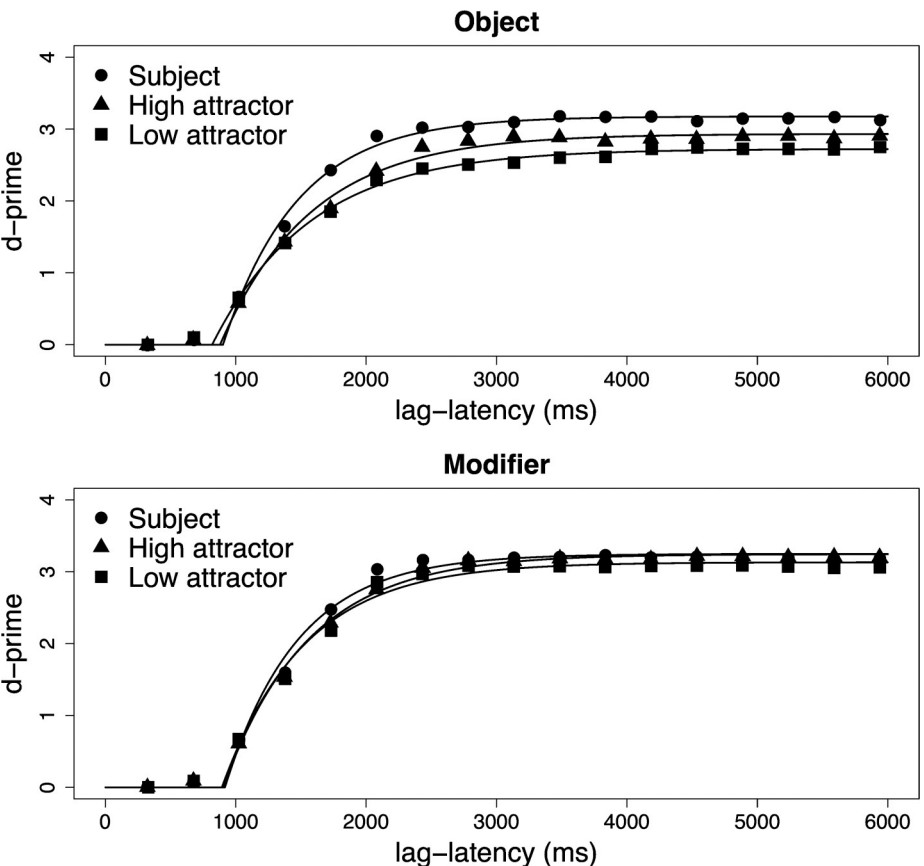

**Fig 3. Probe discrimination as a function of lag-latency.** The visualization of the Lag-latency/d′ series was created by aggregating accuracy data, i.e. first collapsing over subjects, then computing d′. Plot symbols represent empirical d′ values and the smooth lines represent the best-fitting function.

effect of Subject ($\beta_{ProbeWdSubject}$: 0.27, 95% C.I. [0.16, 0.36]; t = 5.3, p < .001); of Structure ($\beta_{Structure}$: -0.22, [-0.31, 0.12]; t = -4.7, p < .001); of Height ($\beta_{Height}$: 0.12, [0.01, 0.23]; t = 2.1, p = .041); and obtained the interactions of Structure and Subject ($\beta_{Strctr \times PrbWdSubj}$: 0.29, [0.10, 0.50]; t = 2.8, p = .005) and of Structure and Height ($\beta_{Strctr \times PrbWdHeight}$: 0.30, [0.08, 0.51]; t = 2.5, p = .013). These coefficients compare closely with the same coefficients from the fit d′ model. The most considerable difference was the estimate of the Height effect: in the regression of the fit d′ reported above, it was smaller ($\beta_{Height}$: 0.09, [-0.02, 0.22]; t = 1.5, p = 0.13). On average, however, fit d′ values varied from empirical d′ values about 1.4%; and a linear regression of the difference between fit and empirical d′ on the experimental coefficients showed no differences. Therefore we conclude that the differences in ultimate attained accuracy, as estimated either by fitting an SAT curve or by smoothing the last four responses, are veridical.

Finally, we turn to participants' performance in the secondary grammaticality judgment task (made after each SAT probe recognition judgment). While we had no specific predictions for this task, the results showed that participants could discriminate grammatical from ungrammatical jabberwocky sentences. They attained accuracy, on average, of 80% for grammatical sentences (the critical SAT trials) and 71% for ungrammatical sentences (the SAT fillers). Table 7 further breaks down accuracy on the grammatical trials by Structure, Probe Status (Target, Distractor), and Probe Word (Subject, High, Low). We computed a mixed-effects logistic regression of grammaticality judgment accuracy on the factors Structure, Probe

**Table 7. Accuracy in post-probe recognition grammaticality judgment task.**

| | | Probe | | |
|---|---|---|---|---|
| Structure | Probe Status | Subject | High | Low |
| Modifier | Target | 83 (8) | 76 (6) | 77 (9) |
| | Distractor | 77 (7) | 72 (7) | 74 (8) |
| Object | Target | 88 (6) | 82 (6) | 82 (4) |
| | Distractor | 82 (4) | 83 (8) | 79 (7) |

Percentage of correct responses in the grammaticality judgment task from Experiment 2 (completed after the SAT probe recognition judgment). Standard error of the mean, over items, is reported in parentheses.

Word & Probe Status (with the same contrast coding as above; Probe Status was sum coded, with Present: +1/2). In accuracy, we found higher performance for Object than Modifier conditions ($\beta_0$: 1.74; $\beta_{Structure}$: 0.38, 95% C.I. [0.30, 0.47]; z = 8.8, p < .001), for Subject than Non-subject conditions ($\beta_{PrbWdSubj}$: 0.28, [0.19, 0.37]; z = 5.9, p < .001), and for Targets than Distractor conditions ($\beta_{Status}$: 0.18, [0.10, 0.27]; z = 4.2, p < .001). These effects were qualified by 2 interactions: (i) Structure and Probe Word Height conditions ($\beta_{Strct\times PrbWdHeight}$: 0.19, [-0.01, 0.39]; z = 1.8, p = .067), reflecting slightly better accuracy (~2%) in High compared to Low conditions for Object conditions, but vice versa for Movement conditions; and (ii) an interaction between Probe Status and Subjecthood ($\beta_{Status\times PrbWdSubj}$: 0.31, [0.13, 0.50]; z = 3.3, p < .001), reflecting the fact that Probe Status had its strongest effect when the probe was the Subject (86% present vs. 81% absent).

Although we are hesitant to read too much into performance in this task, given how demanding the previous SAT phase was, it is notable that participants could nonetheless discriminate grammatical jabberwocky sentences from ungrammatical ones. And, crucially, when they were probed for the subject head noun in the probe recognition, their performance improved on grammaticality judgment. This gives us some added confidence that the same syntactic representation was used to complete the two tasks.

## Discussion

The combined probe recognition and SAT procedure has provided us with two sets of findings with regard to the workings of memory for sentences and to the role of memory in attraction. We discuss them in turns.

Three major data points show important differences between memory for units involved in sentences and memory units involved in lists, suggesting that sentence structure plays a key role in regulating accessibility in memory. First, we found that subject heads are more accessible than any other NP in the sentence. This is true even when the subject was maximally distant from the probe in the linear string, as was the case in sentences with two modifying PPs. Second, we found that the c-commanding object head was more accessible than its modifier, again, despite being linearly further from the probe. These two findings contrast with results from list memorization, where accessibility generally decreases with distance from probe [32]. One may argue that the higher accessibility of the c-commanding attractor as compared to the lower one is due to its position at the beginning of the sentence. Such a possibility is unlikely, for two reasons: (i) the primacy advantage for the first position–as detected by asymptotic accuracy—is typically a weak effect in SAT, compared to the recency advantage [35], [41], [32]; (ii) in line with these previous reports, our findings show that the accessibility of subjects situated in first position in the sentence (in the Modifier condition) does not differ from that of subjects situated further in the sentence, i.e., before the verb (in the Object condition).

Finally, we found that subject heads are systematically retrieved faster than the other NPs from the sentence, independently of their linear position in the sentence. This finding again shows that the dynamics of word retrieval within the sentence differs from that in word lists, where it is usually the final item in the list that enjoys a retrieval speed advantage. It suggests that storage and retrieval from sentences are driven by constraints from the grammar, and in particular here, the subject status and the c-command/precedence distinction.

The main goal of Experiment 2 was to explore the possible alignment between attraction patterns and memory. The data suggest that such an alignment indeed exists. First, subject heads are more accessible and more quickly accessed than the two attractors and, as Experiment 1 demonstrates, participants give the correct grammaticality judgment most of the time (82%-99%, across conditions). Second, we found significantly stronger attraction in Experiment 1 for a c-commanding attractor in Object conditions, compared to the merely preceding one. This asymmetry was reflected in memory accessibility, as measured by asymptotic d' in Experiment 2: the c-commanding attractor was significantly more accessible than the non-commanding attractor in Object conditions. Third, the lack of difference in attraction between the two attractors in the Modifier conditions aligns with the lack of difference found in terms of their accessibility and their retrieval dynamics. More generally, the three attractors that were found to trigger the strongest attraction (the high, c-commanding object attractor and both modifier attractors) showed the highest accessibility in memory, and significantly higher accessibility than the weakest attractor (the low object attractor).

## General discussion

Many authors have proposed that an associative memory plays a key role in attraction, and computational modeling based on ACT-R and related frameworks has shown that such a model captures data [87], [14], [21]. However, the present study is the first that explicitly measured memory retrieval of the critical NPs of the sentence, and examined the alignment between these measures and attraction. The methods used in the two experiments reported is innovative in two respects: we used jabberwocky materials to reduce semantic influences and maximize the role of structural information in performing the task, and we used a direct probe recognition task combined with a speed-accuracy trade-off procedure. Experiment 1 revealed that attraction arises in jabberwocky sentences, and is globally similar in range to that found in natural French sentences [11], [9], [12]: attraction arises in about 15% of trials with the head of a moved preverbal object and with the highest PP modifier situated within the subject phrase. The first experiment also revealed that attraction in jabberwocky sentences is sensitive to the structural distinction between c-command and linear intervention, again replicating data collected with similar structures in natural sentences [12]. Yet, attraction in jabberwocky sentences with double PP modifiers contrasted with findings on the corresponding natural stimuli [6]. Both modifiers were found to trigger similar attraction in jabberwocky, while attraction with the lower PP in natural sentences is significantly weaker.

By specifically probing the various noun phrases of the just-processed sentences, Experiment 2 allowed us to identify differences in their level of accessibility in memory at the precise time point where the critical verb is encountered. Two key findings emerged from that experiment. First, memory accessibility closely aligns with attraction patterns: elements that triggered more attraction in Experiment 1 showed higher levels of accessibility in Experiment 2. Subjects, with which agreement is most of the time correctly realized, are retrieved better and faster; c-commanding object heads, which generate more attraction than their modifiers, are retrieved better; the two PP subject modifiers, which generate similar attraction, give rise to similar accessibility in memory. Second, in contrast to reports from word lists experiments,

differences in memory parameters (level of accessibility and dynamics) cannot be accounted for by the overall recency of the noun phrase in the sentence: subjects are overall retrieved better and faster than any other noun phrase in the sentence, independently of their linear position in the sentence, and noun phrases situated in a position of c-command to the verb are retrieved better than those situated in a position of precedence to the verb, despite being linearly further from it.

In the remainder of the discussion, we discuss, in turns, our view of the memory architecture that underlies attraction, the special status of subjects, the interplay between semantics and syntax in memory, and task effects in attraction in sentence comprehension.

## Hierarchical memory architecture underlying attraction

In the introduction, we reviewed evidence that memory retrieval, for lists and for sentences, is content-addressable. Our finding from Experiment 2 that the strength of attraction coincides with the level of accessibility of the attractor, and not with its retrieval speed, is in line with the hypothesis that the influence exerted by an attractor on sentence processing lies in a content-addressable mechanism relying on cues, rather than on a mechanism relying on search [32]. Additional evidence for content-addressability would come from the observation that attraction is sensitive to similarity. It is mostly in the literature on sentence comprehension that this proposal has been developed, in order to account for the observation that participants read an ungrammatical verb faster if the sentence contains an attractor word that bears the same features than if it does not [21],; [23], [22], [14]. As initially proposed by Wagers et al. [14], this 'grammatical illusion' can be explained by the involvement of a cue-based retrieval process triggered at the verb by agreement marking: the presence of an attractor matching with the verb sometimes gives rise to the erroneous retrieval of that element, satisfying the parser and allowing it to move on faster than if no element matches the verb (see section on Task effects on attraction in sentence comprehension for a discussion of how cue-based retrieval may also be at play in grammatical sentences). Additional evidence comes from a few studies showing that the comprehension of grammatical sentences is also facilitated if the sentence contains an attractor word that bears a different feature from the verb, and thus, from the subject head ([12] [24], [25] but see [88] and for similar evidence in children see [26], [27]).

Interestingly, although attraction in sentence production has traditionally not been interpreted as evidence for the involvement of a content-addressable memory system (but see [87]), a wide array of observations actually seems to attest to the role of the similarity between the agreement controller and the attractor. It is important to note that similarity effects in sentence production contrast with those reported in sentence comprehension in that they cannot show up in terms of agreement feature similarity, since agreement features are not available on the verb. Yet, they do manifest in terms of various morphological, semantic and syntactic features [29]. Morphological case similarity is probably the most prominent factor that has been identified: indeed, most attraction effects reported throughout the production literature actually arise when the subject and the attractor lack case marking, and are therefore non distinguishable with respect to case, either because the language does not express morphological case on the attractor (English, but also French, Spanish, Italian), or because there is a case syncretism [13], [89–91]. In contrast, attraction is virtually nonexistent when the head and attractor have distinct morphological case markers [13], [92–94]). With respect to semantic similarity, some studies have shown that attractors with a high overlap of semantic features with the subject head, in terms of animacy or in terms of semantic field (e.g., *The canoe by the sailboats*) trigger more attraction than those with a lower overlap (e.g., *The canoe by the cabins*, [67], [31]).

Of particular interest to the present study, data showing modulations of attraction due to the syntactic position of the attractor may also be taken as evidence of syntactic similarity-based interference. The most striking finding is that an attractor in a c-commanding position with respect to the verb triggers more attraction than those in a mere precedence position. For example, attraction from the accusative object clitic in French (*Le professeur les lisent*, *The teacher them-ACC read) is stronger than attraction from the dative clitic (*Le professeur leur lit*, *The teacher to them-DAT read, [9]). Similar results were found in Persian speakers [81] who produced significantly more attraction with preverbal accusative objects (e.g., *Parastar chand ta mariz-RA didand*, *The nurse several patients-RA saw-PL) than with preverbal datives (*Parastar be chand ta mariz-RA komak kardand*, *The nurse several patients-RA helped-PL). The same effect was found for sentence-initial objects, with more attraction for accusatives than datives. Whereas the accusative c-commands the verb, the dative is embedded within a (sometimes covert) prepositional layer when it intervenes on the agreement dependency, and thus is only in a position of precedence. Data in French contrasting the dative clitic and the preverbal PP modifier, which also intervenes in terms of precedence, show similar attraction for the two structures [9]. Along the same lines, attraction from the head of a moved complex object c-commanding the verb is stronger than attraction from the same lexical element when it is in a position of modifier of the head [12], a finding which we replicated in Jabberwocky sentences in Experiment 1 here.

C-commanding the verb is a typical property of subjects; it is therefore tempting to propose that the stronger attraction power of c-commanding positions is an instance of syntactic similarity effect, because c-commanding attractors are, at the syntactic level, more similar to subjects. Yet, c-command differs from morphological case and semantic features in that it constitutes a relational property, which is independent of the content of the element itself. But notice that the same issue may arise for how to treat the relational property 'being a subject of' [59], [58], [61], an issue we return to below. Results from Experiment 2 have shown that c-commanding attractors have higher accessibility than those that linearly precede the verb, although their dynamics does not differ, again, a finding that was expected if cue-based retrieval shapes this syntactic modulation on attraction. If the syntactic influence was drawn by properties of a search mechanism, e.g., under the assumption that the c-commanding domain of the verb would be searched first, the difference between the c-commanding object head and its modifier should have become manifest in terms of faster retrieval of the former. The question is whether there is a reasonable way to account for the effect in a content-addressable memory, short of actually implementing a search algorithm. Kush et al. [95], who provide evidence that relational information is used to guide retrieval in the processing of bound variable pronouns, consider this issue in great detail. There are two general (search-free) possibilities: either the features used at encoding must be enriched to encode syntactic domains, or c-command can be "spoofed" by properties of activation. Wagers [96] discusses the possibility that items within the c-command domain of particular heads (like wh-phrases) could have a shared ID or similar context features [97]. If this ID is incorporated into the set of retrieval cues, the mechanism responsible to retrieve the agreement controller is expected to be more prone to erroneously retrieve an intervening element carrying that feature, making it more similar to the target.

Most of the literature on the role of memory in sentence processing has focused on the retrieval of arguments distant from their verb. However, we reviewed in the introduction recent evidence suggesting that memory encoding processes are also involved in agreement processing, as attested by reports showing that gender and number similarity between the controller and an attractor affects sentence processing even if those features are not represented on the verb [29], [24]. The two experiments we reported here were not designed to determine

whether the influence of memory on attraction arises during encoding, retrieval or both. Our finding that the memory parameter that aligns with attraction rates is attractors' accessibility, and not the speed with which they are retrieved, is compatible with the hypothesis that stronger attractors are better encoded, as well as with the hypothesis that they are better retrieved.

Although more work is necessary to specify the mechanisms by which relational information is encoded in memory, as well as the precise locus of the memory influence on attraction, our results provide new, direct evidence that memory representations are shaped by hierarchical structure and key relational constructs (subjecthood and c-command) assumed by syntactic theory.

## The special status of subjects

Results from the SAT experiment show that subjects are more accessible and retrieved faster than any other element in the sentence, and that these two parameters are independent of whether the subject is linearly close to the probe (as in complex object structures) or farther away (as in double PP structures). This latter finding aligns with McElree & Wagers [98] who found that subjects separated from the verb by a prepositional phrase modifier (e.g., *The editor of the journal laughed*) are retrieved as quickly as adjacent subjects. In list memorization SAT measures, accessibility is typically found to decrease with linear distance from the probe, while retrieval dynamics is identical for all elements, except the last one preceding the probe, being faster ([35], but see [19], and [21], for reports of dynamics variations). Our finding, together with that of McElree & Wagers [98], suggest that subjects remain in focal attention, even if they are separated from their verb, at least when the interpolated materials consist of PP modifiers. Our finding that subject probes led to an advantage of approximately 108 ms closely aligns to estimates made for the focus of attention advantage obtained by McElree, Foraker & Dyer [19], who used single-response SAT.

In our materials, subjects always did occupy a special position in the linear string: in the Modifier conditions, it occurred at the beginning of the sentence, while in the complex Object conditions, it occurred at the end. And both the beginning and the end of a sequence are privileged positions. Various studies have shown an advantage for the first-mentioned participant in the sentence [99], [100]. The first content word in the sentence tends to be read more slowly, all things being equal [101], and Gernsbacher suggested that comprehenders use the initial word of the sentence, which is often the subject, to lay the foundation of their mental model, over which upcoming information will be anchored. However, observations show that the first-mention advantage persists even if the second-mentioned element is also part of the subject (as in *Tina and Lisa argued (. . .)*) and if the first element is not the subject (as in *Because of Tina, Lisa was evicted (. . .)*) [100]. Evidence also suggests that sentence comprehension is sensitive to recency effects. When participants hear or read a two-clause sentence, words from the most recent clause are more accessible than those from the earlier clause [102], [103], [99], [104]. However, the timing of the memory task is critical: if the probe is presented together with the last word of the sentence, an advantage is found for the words from the recent clause, but 1400 ms later, an advantage is found for the first-mentioned word, which is arguably reactivated as part of sentence wrap-up processes. Could the faster dynamics and higher accessibility of subjects in our study thus be explained in terms of the primacy and recency effects? While primacy and recency may have contributed to the accessibility of the nouns in our study, recall that we always found a dynamics advantage for the subject head, even when it was not the first word in the sentence. Moreover, previous SAT studies showed that the recency advantage shows up in the asymptote, but not in the dynamical parameters[32].

What is it that makes subjects special? One possibility is that the subject function is intimately linked to the construct of 'actor' or 'agent', which is rooted in our ability to understand goal-directed actions: "Agents are a class of objects possessing sets of causal properties that distinguish them from other physical objects (. . .) as a result of evolution, we have become adapted to track these sets of properties and to efficiently learn to interpret the behavior of these objects in specific ways" [105]. Evidence indeed suggests that the human attention system has a special sensitivity to tracking humans and animals, which are good potential actors [106]. The extended Argument Dependency Model [107] assumes that an actor-based principle guides sentence comprehension, by which the system is primarily designed for seeking to identify the actor, that is, the participant responsible for the state of affairs expressed in the sentence. The linguistic features related to actor identification are: +self (under the view that the first person is the prototypical actor), +animate/human, +definite/specific, +first position, +nominative. Such a model is compatible with content-addressable memory retrieval models, in which actor identification is sensitive to competition from candidates with overlapping features although it involves specific assumptions about the weighing of the cues, which may vary cross-linguistically [107], [108]. An important property of the model is that interference is 'actor-centered' in that only features of actors are relevant in generating similarity-based interference [109]. However, our results show that the subjects remain privileged in memory even in the absence of semantics. Arnett & Wagers [61] argued that subject phrases are directly encoded for their case or syntactic position in the phrase structure tree. Building from the design of Van Dyke & Lewis [60], they showed that when participants read finite clauses, the only kinds of subjects that could interfere were those that were the subjects of other finite clauses. A variety of other subject properties (e.g., being the agent in an event nominalization: "the marauder's destruction of the village") were ineffective at generating similarity-based interference.

Subjects have many properties that should give them a survival advantage in memory, such as the fact that they often occur first and that they often name the actor participant in an event. But our results show that, even when neither of those properties are relevant, subjects retain their special status in memory. The most plausible remaining explanation, in our view, is that this status derives from their prominent structural position in our stimuli. In our task, the subject phrase was always the last NP comprehenders encountered that c-commanded the verb. If our sentences had more elaborate complements, it may have been other NPs beside the subject would show a similar advantage. Future research is needed to explore such possibilities.

Finally, our results also suggest that elements from the subject constituent may have a special role to play in attraction, as subject modifiers seemed to be more activated than any of the elements from the object constituent; in fact, their asymptote was nearly as high as that of subjects (although a clear difference remained in their dynamics). Using a two-choice response time paradigm, Staub [110] found that although response times for correct agreement decisions were similar for structures with subject modifiers and those with moved objects, their underlying distribution was qualitatively different: whereas the effect of a plural subject modifier is due to both a shifting of the distribution to the right and to increased skewing, the effect of a plural object is almost entirely due to skewing. This led him to conclude that different mechanisms underlie attraction in these two structures (in line with the early proposal by [2]). Although our results also suggest qualitative differences between the two types of elements with respect to their memory status, it is important to keep in mind that both our data and Staub's rely on comparisons across sentences with different structures. Hence, further research is needed to determine whether the difference holds within structures containing both a moved object and PP modifiers, and if so, to better understand whether the difference lies in the sentence processing mechanism underlying these two types of elements, their memory

status, or in the linking between memory and processing mechanisms, which remains to be fully fleshed out.

## The interplay of semantics and syntax in memory

An important part of the literature on attraction has been concerned with the influence of semantics. Most of the research has focused on the role of the notional representation of the subject phrase [64], [65], [31], [66], and on the influence of semantic correlates of grammatical number and gender features [111], [112]. A few studies also explored the influence of the semantic relationship between the head and the attractor noun, and showed influences on agreement from the semantic similarity between the subject head and the attractor [67] and from the semantic integration between the head and the attractor [113], [7]. However, most of the studies on attraction failed to control for these semantic relations within the sentence. The use of jabberwocky materials is a first attempt to explore attraction while controlling it. The comparison between previous results on natural stimuli and the current results on similar structures with reduced semantics provides us with new insights about the role of semantics in attraction.

In line with natural language data on object attraction [12], we found that hierarchical height, which coincided with c-command, significantly affected attraction in jabberwocky sentences involving a moved complex object: grammaticality judgments were more penalized (both in terms of accuracy and RTs) when the c-commanding attractor, i.e., the object head, mismatched the subject's number than when the lower attractor, i.e., the object's modifier, mismatched it. Results from the SAT experiment also showed a significant difference in the memory availability of the two nouns, aligning with the attraction profile. This finding suggests that this effect, lying in the structural difference between c-command and precedence, is independent of the semantics. However, a different profile emerged for double PP structures. In contrast to natural language data on sentences with two PP subject modifiers, which showed virtually no attraction from the lower attractor [6], [7], no difference was found between the two attractors with jabberwocky sentences: both generated significant attraction in grammaticality judgements, and both were equally available in memory. Although one cannot exclude the possibility that the lack of a height effect in jabberwocky double modifiers is due to lack of power, the fact that it was systematically found across the two experiments, in contrast to the significant effect found in object modifiers, calls for explanation. How can we account for the difference between jabberwocky and natural sentences in double modifier sentences?

A first difference between natural and jabberwocky data lies in the task procedure: whereas the previous findings from natural language sentences were obtained in sentence completion tasks, our current finding for jabberwocky is from a grammaticality judgment task. However, the task does not appear to be responsible for the difference: grammaticality judgments on natural French sentences were shown to give rise to stronger attraction with the highest PP [114], thus replicating results obtained with a sentence completion procedure. Independent sets of experimental evidence on other structures have shown that grammaticality judgments consistently replicate structural modulations of attraction found with the sentence completion task, suggesting that the two tap into the same mechanism of structure building [12], [71]

Another key difference between jabberwocky and natural language data lies in the absence of lexical semantics in the former. As a result, two syntactically correct structures can be built in double PP conditions: one in which the second PP is adjoined to the entire subject phrase, and one in which it is nested inside the first PP. Semantics in the materials from Franck et al. [6] promoted the nested type, so that the second PP was embedded in the first PP. The lack of semantic information in the jabberwocky materials may have promoted more interpretations

of the first type, in which the second PP modifies the subject directly. If this were the case, both PPs would be at the same syntactic distance to the head, which would explain the lack of a height effect. Nevertheless, results from Gillespie and Pearlmutter [7] potentially challenge this hypothesis. The authors contrasted sentences similar to Franck et al. [6] with a nested hierarchical structure, in which the second PP modified the first PP (e.g., "The backpack with the plastic buckles on the leather strap"), to sentences with a flat structure in which both PPs modified the head (e.g., "The highway to the western suburbs with the steel guardrail"). Results showed that the increase of attraction for the highest, first PP was actually independent of whether the second PP is attached to the first PP or to the head: both embedding and flat structures showed stronger attraction with the first PP. In another experiment, the authors manipulated the semantic integration between the PP and the head noun: for example, if one compares the phrases "the book with the torn pages" and "the book by the red pen", the PP *with the torn pages* is more closely integrated to the head *the book* than the PP *by the red pen*. Gillespie and Pearlmutter found stronger attraction from the higher PP than from the lower when the former was highly semantically integrated to the head (e.g., *The book with the torn pages by the red pen* generated more attraction than *The book with the torn page by the red pens*), but no difference between the two PPs when the highly integrated PP is the lower PP (e.g., *The book by the red pens with the torn page* generated similar attraction to *The book by the red pens with the torn page*). The authors concluded that attraction rates were determined by a combination of linear distance to the head and semantic integration. They argued that the mechanism underlying the effect of these two factors is sentence planning: when the attractor is linearly closer to the head and/or when it is more integrated semantically to it, it tends to be 'planned' at the same time as the head; in that case, the two nouns are simultaneously active in memory and their numbers therefore have increased chance to interfere. In this so-called 'Scope of planning' account, timing of planning is primarily determined by the linear order in which elements have to be produced: hierarchically higher PPs generate more attraction not in virtue of their hierarchical height but in virtue of their linear closeness to the head. However, semantic integration has the potential to shift the planning (in line with [113]): elements that are linearly distant but semantically close to the head may still be planned at the same time, and thus have an influence on attraction.

Nevertheless, our new findings on attraction in jabberwocky sentences, in which semantics plays no role and the only factor at play is thus linear proximity, do not support the Scope of planning account. In jabberwocky sentences, the Scope of planning account predicts that linear distance to the head will be the only determinant of attraction, since semantic integration is switched off. Hence, in our double modifier structures, the first PP being closer to the head should generate stronger attraction than the second PP. Our data yielded no evidence that linear closeness to the head increases attraction. Also, the Scope of planning account predicts that the modifier PP in our complex object structures should trigger more attraction than the object head, because it is linearly closer to the subject head. Our data showed the opposite: the object head generates more attraction than its PP modifier, despite being linearly further from the subject head.

In sum, neither specificities of the task, nor distance (hierarchical or linear), nor semantic integration seem to capture the full set of observations on natural and jabberwocky sentences with two PPs. In particular, our data suggest that the lack of semantics in jabberwocky sentences modified the attraction effect found for natural sentences with double modifiers, but not those with complex objects. Hence, assuming that the lack of a difference between the two PPs in the jabberwocky testing of double modifiers is not due to lack of statistical power, semantics had a different impact on these two structures. Why would semantics have a different impact on these two structures? The key difference between the two structures is that

whereas height is tied to the structural relation of c-command in complex objects, it is looser in the case of PPs. As we already discussed, double PP sentences can have two different underlying structures: an embedded structure or a flat structure. But even single PP sentences may have different underlying structures, as subject modifiers have the option of being arguments or adjuncts [115]. Although these factors may not play a significant role in attraction per se [113], we suggest that the unconstrained syntactic structure of the double PP structure is responsible for maintaining the two PPs in a similar memory state, and thus give them a similar attraction potential. We propose that semantics would play a key role in constraining syntactic structure building and thus providing a stable representation to store in memory when structural constraints are undetermined. In the absence of semantics, double PP sentences would remain 'floating' without finding a clear memory anchor, giving rise to similar attraction from the two PPs, in contrast to natural sentences. In contrast, for complex objects, there is no other alternative parse tree: the object is necessarily an argument of the verb, and the first NP is necessarily the head of that complex constituent. Semantics is unnecessary for building the underlying hierarchical structure, which would be why similar, structure-based results are found for both jabberwocky and natural sentences.

## Conclusion

This study presents two new tools providing new avenues for the study of the relations between sentence processing and memory: the use of jabberwocky materials allowed us for maximally controlling the role of semantics and lexical frequency, and thus focus on the specificities of syntactic processing, while the transposition of the probe recognition task to sentences with a Speed-Accuracy Trade-off design allowed us exploring the structure of memory representations underlying sentences. The data show that attraction arises independently of semantics, under the guidance of structural principles, similar to attraction in natural sentences. More critically, the data provide direct evidence for the role of memory as a key factor in agreement and attraction, in that more accessible elements in memory generate more attraction. It suggests a tight link between theoretical constructs from syntactic theory, in particular those of subject and c-command, and memory representations, and with this, suggests that the theory of linguistic competence is an important component of the theory of linguistic performance.

## Acknowledgments

We wish to thank Brian Dillon, Brian McElree, Whit Tabor and Julie Van Dyke for enriching discussions on this work. Nevertheless, we take complete responsibility for the content of the paper.

## Author Contributions

**Conceptualization:** Julie Franck.

**Data curation:** Julie Franck.

**Formal analysis:** Matthew Wagers.

**Funding acquisition:** Julie Franck.

**Investigation:** Julie Franck.

**Methodology:** Julie Franck.

**Project administration:** Julie Franck.

**Software:** Matthew Wagers.

**Supervision:** Julie Franck.

**Visualization:** Matthew Wagers.

**Writing – original draft:** Julie Franck, Matthew Wagers.

**Writing – review & editing:** Julie Franck, Matthew Wagers.

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
