## [Decision Letter · Decision Letter 0]

13 Nov 2019

Re: Submission PONE-D-19-23548 “Hierarchical structure and memory retrieval mechanisms in agreement attraction” by Julie Frank & Matthew Wagers

Dear authors,

Thank you for submitting your work to PLOS ONE. I have now received reviews from two experts in the field. As you will be able to see, both of them recommend some relatively major modifications to your manuscript. In general, I agree with the reviewers that your work may eventually be published in PLOS ONE. I would also like to encourage you to thoroughly address the reviewers’ comments in your revised manuscript. My personal reading of your manuscript is largely congruent with both reviewers’ comments. Please note that this invitation to submit a revised version of your work does not guarantee eventual publication in PLOS ONE.

Kind regards,

Andriy Myachykov

Journal Requirements:

3. Please remove your figures from within your manuscript file, leaving only the individual TIFF/EPS image files, uploaded separately.  These will be automatically included in the reviewers’ PDF.

Reviewers' comments:

Reviewer's Responses to Questions

**Comments to the Author**

1. Is the manuscript technically sound, and do the data support the conclusions?

Reviewer #1: Partly

Reviewer #2: Yes

2. Has the statistical analysis been performed appropriately and rigorously? 

Reviewer #1: Yes

Reviewer #2: Yes

3. Have the authors made all data underlying the findings in their manuscript fully available?

Reviewer #1: Yes

Reviewer #2: Yes

4. Is the manuscript presented in an intelligible fashion and written in standard English?

Reviewer #1: Yes

Reviewer #2: Yes

5. Review Comments to the Author

Reviewer #1: Summary of the research and overall impression

The authors report the results of two experimental studies which investigated the role of structural factors in agreement attraction, and whether agreement attraction is linked to memory retrieval operations. The first study used speeded grammaticality judgments of Jabberwocky sentences in which all content words except the verb were replaced with nonce words. Results show that agreement attraction arises in such sentences, where semantic influences are eliminated, that the head noun of a preposed object causes more attraction than a noun contained in a modifier within the the same NP, and no evidence that structural position matters within PP modifiers of the subject. The second study used a speed-accuracy tradeoff procedure with a probe recognition task and grammaticality judgments. Results (in terms of asymptotic accuracy) show that subjects are easier to recognize than non-subjects, that heads of preposed objects are easier to recognize than non-heads, and that both nouns in complex PP modifiers of the subject are equally easy to recognize, almost on par with the subject. The authors argue that taken together, the results of the two studies support the notion that agreement attraction is due to cue-based retrieval from memory, and that structural factors (specifically, c-command and subjecthood) influence memory retrieval.

My overall impression is that the reported studies are for the most part methodologically sound and that the statistical analyses were conducted correctly (with some criticisms, see comments below). However, there are some very important concerns about statistical power (which have already been mentioned by a previous reviewer) and about the interpretation of the results which I would like to see addressed in a revision. I would also suggest changing the structure of the paper somewhat in order to make it more readable. Nevertheless, due to both the novel nature of the stimuli (Jabberwocky) and the use of the SAT method, this work definitely contributes a lot to the theoretical debate about agreement attraction.

Major comments

One major issue is that the authors do not give enough room to "encoding-based" accounts of agreement attraction: The authors mention the Marking and Morphing model in the introduction, but afterwards focus solely on a retrieval-based view of the phenomenon. I believe the combined findings from the grammaticality judgment study and the SAT task are equally compatible with the view that the encoding of the subject's number feature is contaminated during the encoding of the other nouns in the sentence, and that this encoding interference is sensitive to structural factors. The authors should definitely address this possibility, even if they have reasons to believe that it's not the right explanation (they dismiss Marking and Morphing as being unable to explain the grammatical asymmetry, but Hammer et al. have recently argued that it may be due to response bias).

The paper is somewhat difficult to read. The "Overview of the study" section is very long and discusses both studies, one after the other, along with the authors' predictions, which are then followed by the experiments themselves. I believe it would make more sense to reorganize the paper so that an overview of Experiment 1 is first given, followed by the description of Experiment 1, its results and their discussion, followed by the overview of Experiment 2, and so forth. As it is, by the time the reader has digested the information about SAT in the overview section, they will have forgotten the predictions for Experiment 1. It would also be nice to have a table summarizing the predictions for each of the experiments in a concise way. To increase comprehensibility, it would also be nice to be given an overview of the conditions, along with examples, for Experiment 2. I would suggest also incorporating into the examples which NP each probe type in the SAT task would correspond to.

Like another reviewer before me, I am not convinced by the authors' use of an "Attraction Index". One reason is that it effectively removes the uncertainty involved in measuring the effect of Match by averaging over all trials for each item. Reporting the results of the logistic regression is reassuring in the sense that this uncertainty is taken into account, but I would suggest removing the "Attraction Index" from the manuscript entirely - especially as the results section of Experiment 1 reports logistic regression results before immediately switching to "Attraction Index" results, which is confusing to the reader. One page 21, "Attraction Index results on accuracy" are mentioned, which is equally confusing.

The reported results from Experiment 1 with regard to judgment RTs were confusing to me and did not seem to be discussed in any detail. On page 24, the manuscript says that "Low Object conditions showed a greater Match effect", but the reported result should refer to the Low Modifier, as the paragraph talks specifically about the Modifier conditions only. There is apparently a structural or linear effect here that is not reflected in the accuracy measures, and this fact should be discussed in the manuscript.

Connected to the previous point, there is a general issue of the design and interpretation of the studies being overly complex. Experiment 1 manipulates four experimental factors across two measures, and the fitted statistical models contain all the possible interactions. Combined with the low power (small sample size) of the studies, there are issues concerning both Type I and Type II error here, such that *any* effect being significant is highly likely given the large number of comparisons, and that an effect *not* being significant does not signal the absence of an underlying true difference between conditions. The authors should explicitly acknowledge these issues in the paper.

I do not understand the point made on page 26 about the difference between reading for comprehension and grammaticality judgments. Specifically, for grammaticality judgments, I fail to see how "the agreement features of the verb, which need to be checked, cannot be used as retrieval cues". To me, it is perfectly sensible to assume that the verb triggers a retrieval for an agreement-licensing subject in these cases. Also, in their response to Reviewer #1, the authors say that "the

misidentification of the attractor as being the agreement controller can only lead to a detectable error when the attractor mismatches the controller" - while this is true, it appears to me that it can be seen as a flaw of the task, given that there may be errors that are not identifiable as such (wrong agreement controller with correct feature). I believe this point should be acknowledged.

The authors argue that the results of Experiment 1 and Experiment 2 align, and that probe recognition performance maps onto agreement attraction strength. While this interpretation is generally sensible in light of the results, retrieval in sentence comprehension appears to be more akin to a recall than a recognition task. I would like the authors to comment on this, as well as on the fact that recognition dues not necessarily involve matching of any grammatical cues. Furthermore, it is somewhat troubling to me that the results of the grammaticality judgment task of Experiment 2 were not analyzed - can the authors justify this decision, given that it would be good to see a replication of the results from Experiment 1?

A previous reviewer took issue with the fact that predictions from a retrieval model based on ACT-R (Lewis & Vasishth, 2005) were conflated with those of non-ACT-R models. The authors apparently responded to this criticism by removing references to ACT-R from the manuscript. Like the previous reviewer, I would like the authors to instead discuss the predictions of the different models with regard to processing speed in the SAT task, as the ACT-R model predicts that speed *should* be affected by the agreement attraction manipulation.

In their response to Reviewer #2, the authors explain why they do not think that the claimed effect of c-command reduces to an effect of primacy. Yet, in the manuscript proper, it is simply stated on page 32 that "we do not consider the explanation very likely given the strength of the effect". I think the full explanation should be given in the manuscript.

Minor points

On page 8, "The asymptote provides a measure of the overall probability of correct retrieval" should be changed to "maximum probability" if I understand the reasoning behind SAT correctly.

Also on page 8, "the asymptote decreases gradually with recency of list position" - I believe the authors mean "increases", looking at McErlee & Dosher (1989). The authors make the same claim again on page 39 ("accessibility is typically found to increase with linear distance to the probe"), so maybe I am misunderstanding something.

On page 11, Lewis & Vasishth (2005) should be cited as a model that assumes a narrow focus of attention (a single-element buffer).

On page 20, the authors state that 500 ms per region were chosen "to minimize judgment errors". Are agreement attraction errors not judgment errors?

On page 28, I would like to know more about how the sessions were conducted. Did they all take place on one day or were they spread across different days? Especially if they were spread across several days, participants may have developed strategies.

Typo on page 29, "that it *refers* specifically to the x-intercept".

On page 31, I think the authors mean "a significant interaction between Probe type and Structure" rather than "a significant interaction between Subject probes and Structure".

On page 33, I am no sure if something can "be agreed with" something else in standard English.

On page 34, I am not convinced that the Jabberwocky sentences "ensured that the task taps into structural processes" - all they did was ensure that people did not resort to semantics for additional guidance, I think. See the point about c-command being confounded with primacy above.

On page 36, "the presence of an attractor matching with the verb gives rise to the erroneous retrieval of that element, satisfying the parser" - add "sometimes".

On page 36, regarding absence of attraction effects in the presence of case marking: Recent work by Avetisyan et al. (https://osf.io/tpyjm/) suggests that case marking reduces attraction in production, but not necessarily in comprehension.

Reviewer #2: I am a new reviewer in this process and therefore cannot comment directly on the changes made in response to the previous reviews (I haven’t seen the original manuscript). However, my impression from reading the ‘Responses to Reviewers’ document is that the authors took great care in addressing most of the issues previously identified.

The paper reports a scientifically rigorous line of research on agreement attraction. Although I would not consider myself an expert on this topic, I appreciate the novelty of the approach taken: (a) use of jabberwocky sentences in French in order to reduce potential semantic influences on agreement attraction, and (b) use of the response-signal SAT paradigm combined with a memory probe reaction task in Experiment 2, which allows for a clearer distinction between accessibility vs. dynamics in the underlying memory retrieval processes. I find the authors’ interpretation in support of a content-addressable architecture quite convincing. Moreover, the general discussion is very thorough and nuanced. Overall, I think this paper would make a very useful contribution to the literature on agreement attraction.

Apart from a number of line-by-line comments (see further below), there are three more general points for revision that I would like the authors to consider:

(1) Reporting of p-values:

Reporting p-values as “p < .05”, “n.s.” etc. appears rather old-school and is neither very informative nor helpful for potential meta-analyses (p-curve etc.). Throughout the results sections, I would prefer if actual p-values (rounded to 3 decimals) were being reported, both for significant and non-significant effects. The only exception is “p < .001” (i.e., when the actual p-value is smaller than .001).

(2) Discussion of Experiment 1:

The discussion on p. 24-26 does not make clear whether the authors primarily rely on the ‘attraction index’ findings (Fig. 1), the RT findings (Fig. 2), or both, in drawing specific theoretical conclusions from Experiment 1. Generally, I think this section could benefit from a more thorough comparison of the results across the two DVs. For example, it is stated that “… in the double-modifier condition, both attractors generated similar rates of attraction.” (p. 25); while this appears to be true for the ‘attraction index’ data (Fig. 1), the pattern in the RT data (Fig. 2) actually looks quite different, especially for ungrammatical sentences (or am I misinterpreting something here?). Please clarify, as one could otherwise get the impression that potentially ‘inconvenient’ aspects of the results are being ignored.

(3) Bounded exponential curve fitting (Experiment 2):

A full 12 (conditions) * 3 (lambda, beta, gamma) = 36 parameter model has been fitted to the data in Experiment 2. I appreciate that the authors are aware of the problem of parameter trade-off as a result of overfitting (cf. motivation for the 1/beta + delta measure on p. 30), but I’m not quite sure whether the measures taken to mitigate this problem are sufficient. In particular, is it safe to assume that there are no trade-offs between lambda (asymptote) and any of the speed parameters (beta, delta)? Given the large number of conditions, it would seem reasonable to determine the amount of parameter variation /necessary/ to describe the observed cross-condition differences first. This could be done via some backward model selection procedure, which would end up with a more parsimonious fixed-effects model of the data. (NB, such an approach has also been used in many of the McElree et al. papers cited).

Line-by-line comments:

P. 4, second paragraph: There seems to be a noun missing in “For example, attraction was found with plural objects in the production of various involving movement of the object in preverbal position, …”

P. 5, second paragraph: Could you perhaps provide more example sentences here?

P. 6: What is a “clause-mate subject”? Is this an established term?

P. 7, last paragraph: “The response-signal speed-accuracy trade-off (SAT), a procedure…”. Maybe add “paradigm” after “(SAT)”.

P. 12: Maybe provide some ‘double-PP’ examples here as well, not just object question examples.

P. 13, top: Delete “of” in “…which would attest of a direct link between attraction and memory, …”

P. 15: “Here, the use of the probe recognition task provides a purer measure of how well…” – perhaps replace “purer measure” with “more direct measure” (?)

P. 16, third bullet point after equation: The lambda parameter is not “unitless”, is it? It should be in d’ units here (i.e., indexing d’ at t = infinite).

P.20, top: The description is not very transparent in terms of how the sentences were segmented for presentation. Perhaps indicate item segmentation in the examples provided in Table 1 (p. 19), using square brackets or something. NB, 500 ms per segment would hardly qualify as “Rapid” Serial Visual Presentation.

P. 21, second paragraph: What is the motivation behind analysing log RTs? Log-transforming data not only makes the data appear more ‘normal’ (less positively skewed), but also changes theoretical interpretation as it effectively turns a linear model into a multiplicative (or ‘log-linear’) one, which is not always desirable. A potentially better alternative might be to use a Gamma(identity) model family, which assumes a positively skewed data distribution without giving up the assumption of linear relationships between predictors and DV.

P. 21, bottom: “ones swith” -> “ones with”

P. 25, end of second paragraph: “The lack of semantic information in the jabberwocky materials may have promoted more instances of the first type, …” – perhaps replace “instances” with “interpretations” (would sound better in this context).

P. 29, 3rd paragraph: Verb missing in “although note that it specifically to the x-intercept.” (“applies”?)

P. 38: “Yet, c-command differs from morphological case and semantic features in that it lies in a relational property, …” -> “… in that it constitutes a relational property, …” would sound better.

P. 38: “…the difference between the c-commanding object head and its modifier should have manifest…” -> better: “… should have become manifest …”

P. 39, under “The special status of subjects”: Verb missing in “ … independent of whether the subject is linearly to the probe …” (?)

P. 42, second paragraph: “This led him conclude …” -> “This led him to conclude”

P. 43, end of second paragraph: “structures without semantics” is perhaps a bit too strong in relation to the current materials (only the nouns were fictitious, while the verbs clearly had a meaning).

P. 46: Delete “it” in “As we already discussed it, …”

6. PLOS authors have the option to publish the peer review history of their article (what does this mean?). If published, this will include your full peer review and any attached files.

Reviewer #1: No

Reviewer #2: Yes: Christoph Scheepers

---

## [Author Response · Author response to Decision Letter 0]

27 Jan 2020

We have submitted a detailed response to the reviewers in a separate file, which we uploaded on the platform with the rest of the documents.

---

## [Decision Letter · Decision Letter 1]

24 Feb 2020

PONE-D-19-23548R1

Hierarchical structure and memory mechanisms in agreement attraction

PLOS ONE

Dear Dr. Franck,

Thank you for submitting your manuscript to PLOS ONE. After careful consideration, we feel that it has merit but does not fully meet PLOS ONE’s publication criteria as it currently stands. Therefore, we invite you to submit a revised version of the manuscript that addresses the points raised during the review process.

We would appreciate receiving your revised manuscript by Apr 09 2020 11:59PM. To enhance the reproducibility of your results, we recommend that if applicable you deposit your laboratory protocols in protocols.io, where a protocol can be assigned its own identifier (DOI) such that it can be cited independently in the future. For instructions see: http://journals.plos.org/plosone/s/submission-guidelines#loc-laboratory-protocols

We look forward to receiving your revised manuscript.

Kind regards,

Andriy Myachykov, PhD

Academic Editor

PLOS ONE

Reviewers' comments:

Reviewer's Responses to Questions

**Comments to the Author**

1. If the authors have adequately addressed your comments raised in a previous round of review and you feel that this manuscript is now acceptable for publication, you may indicate that here to bypass the “Comments to the Author” section, enter your conflict of interest statement in the “Confidential to Editor” section, and submit your "Accept" recommendation.

Reviewer #1: (No Response)

Reviewer #2: All comments have been addressed

2. Is the manuscript technically sound, and do the data support the conclusions?

Reviewer #1: Partly

Reviewer #2: Yes

3. Has the statistical analysis been performed appropriately and rigorously? 

Reviewer #1: Yes

Reviewer #2: Yes

4. Have the authors made all data underlying the findings in their manuscript fully available?

Reviewer #1: Yes

Reviewer #2: Yes

5. Is the manuscript presented in an intelligible fashion and written in standard English?

Reviewer #1: Yes

Reviewer #2: Yes

6. Review Comments to the Author

Reviewer #1: The manuscript has improved significantly after the previous round of reviews. Nevertheless, I believe some changes are still in order. Detailed comments are found below. While I will not insist on my general request to acknowledge the possibility of statistical errors, the authors should report confidence intervals around the model estimates throughout the manuscript (as recommended, inter alia, by the APA) to allow the reader to gauge the uncertainty of the estimates. Furthermore, the authors use a lot of space to discuss a null result (or two null results, rather - the lack of a difference between the two modifiers in both studies) in what I still suspect to be a relatively low-power situation. Here, the authors should explicitly acknowledge that we could be looking at false negatives. I also suggest shortening the section "The interplay of semantics and syntax in memory", because the differences between Jabberwocky and natural sentences may not be as important as the authors make them out to be. In addition, I fail to see how the "Scope of planning" account of Gillespie and Pearlmutter (2011), which was proposed for production, is relevant to a comprehension study.

Detailed comments:

- In the abstract, "Speakers occasionally cause the verb to agree with an element that is not the subject, a so-called ‘attractor’; likewise, comprehenders occasionally fail to notice agreement errors when the attractor agrees with the verb."

I suggest a slight reformulation:

"Speakers occasionally produce verbs that agree with an element that is not the subject, a so-called ‘attractor’; likewise, comprehenders occasionally fail to notice agreement errors when the verb agrees with the attractor."

- On page 4, are the authors certain that Marking & Morphing fails to predict attraction in contexts that are not the typical NP-PP-(PP) configurations? Eberhard et al. (2005:544) state that "Because SAP [singular and plural] may flow unobstructed throughout a structural network, number information bound anywhere within a structure has the potential to influence agreement processes." If I understand this statement correctly, the NP-PP configuration is not the only one where agreement attraction should occur.

- On page 5, first paragraph: "it is a NP" -> "it is an NP". This pattern also appears in other places throughout the manuscript. I am not a native speaker of English, but I do not resolve the acronym to "noun phrase" when I read the text, and the form of the determiner should thus agree with the pronunciation of the acronym.

- No confidence intervals are reported for the reaction time analysis of Experiment 1, nor is there a table showing results in the same way as for the logistic regression. I would suggest adding this information so that reporting is the same across the different dependent variables. The CIs of the estimates should also be given in the text of the results section Experiment 2, so that the reader can better gauge the uncertainty of the estimates (of the differences between conditions).

- There is a mismatch in the reporting of the RT results from Experiment 1 and the secondary task from Experiment 2 versus all other reported analyses in the sense that the RT/judgment results are reported with "beta(factor) = XX" whereas all other results are reported without any beta being mentioned, giving only bare numbers. The betas should appear consistently across all analyses.

- At the bottom of page 27, and later from page 50 onwards, the authors discuss the mismatch between their current findings and those of previous studies in terms of the absence of a difference between the two modifiers. Here, the possibility should be acknowledged that the lack of a difference in the current study may simply be a statistical false negative result, as opposed to reflecting a theoretically interesting divergence between Jabberwocky and non-Jabberwocky sentences. Looking also at the results of Experiment 2, it is of course perfectly possible that the authors are on the right track with their proposed explanation for the difference, but the more parsimonious explanation would be that the effect was masked by noise and therefore not detected in the current studies. As it is, the authors spend several pages discussing the implications of a null result, even though absence of evidence is not evidence of absence (e.g. Altman & Bland, 1995).

- On page 35, the authors write that "the effect of being a non-subject probe was larger in Object conditions (0.41, t=5.0, p < .001) than in Modifier conditions (0.11, t=1.9, p=.066)". But shouldn't the asymptote be higher for subjects than for non-subjects? So the estimates should either be negative or the sentence should refer to subject probes.

- On page 39, "In line with those previous reports" -> "these previous reports"

- Also on page 39: "The main goal of Experiment 2 was to explore the possible alignment between attraction patterns and memory. The data suggest that this is the case." The referent of "this" is unclear. I suggest changing the second sentence to "... suggest that such an alignment indeed exists".

- On page 51, I do not understand how the Gillespie and Pearlmutter (2011) account is relevant to the current experiments. While it is true that the Scope of planning theory takes semantic integration into account, it is fundamentally a theory of production rather than comprehension. As such, I do not see why the SOP account would predict anything at all for the current experiments, and why the authors argue that their results provide evidence against the SOP account.

- On page 52, the authors are overstating their case when they say that "linear closeness to the head does not increase attraction". This statement should be changed to "Our data yielded no evidence that linear closeness to the head increases attraction". The same is true for the statement about the final sentence of the paragraph, which should be qualified be adding "in our data", at the very least.

- Still on page 52, the final paragraph needs to be qualified by highlighting the speculative nature of the authors' proposed explanation, and taking into account the possibility that the differences in findings are not meaningful at all but simply due to statistical error.

- I fail to understand the final part of the final sentence on page 54. How do the results show that the theory of competence is also a theory of performance? I agree that performance needs to be studied to get a clear picture of what competence is, but one doesn't automatically need to equate the two. This is also the first time in the entire manuscript that the concepts of competence and performance are mentioned. Finally, I am not sure which hypothesis is being "reopened" by the findings - I'm sure that decades of psycholinguistic research were not conducted in the belief that the competence/performance debate had been settled. The authors should clarify what they mean here.

Reviewer #2: I am satisfied with how the authors have responded to my previous comments, which have all been addressed in the revision. I therefore endorse publication.

7. PLOS authors have the option to publish the peer review history of their article (what does this mean?). If published, this will include your full peer review and any attached files.

Reviewer #1: No

Reviewer #2: Yes: Christoph Scheepers

---

## [Decision Letter · Decision Letter 2]

9 Apr 2020

Hierarchical structure and memory mechanisms in agreement attraction

PONE-D-19-23548R2

Dear Dr. Franck,

We are pleased to inform you that your manuscript has been judged scientifically suitable for publication and will be formally accepted for publication once it complies with all outstanding technical requirements.

With kind regards,

Andriy Myachykov, PhD

Academic Editor

PLOS ONE

Additional Editor Comments (optional):

Reviewers' comments:

Reviewer's Responses to Questions

**Comments to the Author**

1. If the authors have adequately addressed your comments raised in a previous round of review and you feel that this manuscript is now acceptable for publication, you may indicate that here to bypass the “Comments to the Author” section, enter your conflict of interest statement in the “Confidential to Editor” section, and submit your "Accept" recommendation.

Reviewer #1: All comments have been addressed

2. Is the manuscript technically sound, and do the data support the conclusions?

Reviewer #1: Yes

3. Has the statistical analysis been performed appropriately and rigorously? 

Reviewer #1: Yes

4. Have the authors made all data underlying the findings in their manuscript fully available?

Reviewer #1: Yes

5. Is the manuscript presented in an intelligible fashion and written in standard English?

Reviewer #1: Yes

6. Review Comments to the Author

Reviewer #1: (No Response)

7. PLOS authors have the option to publish the peer review history of their article (what does this mean?). If published, this will include your full peer review and any attached files.

Reviewer #1: No

---

## [Editor Report · Acceptance letter]

20 Apr 2020

PONE-D-19-23548R2 

Hierarchical structure and memory mechanisms in agreement attraction 

Dear Dr. Franck:

I am pleased to inform you that your manuscript has been deemed suitable for publication in PLOS ONE. Congratulations! Your manuscript is now with our production department. 

With kind regards,

on behalf of

Dr. Andriy Myachykov 

Academic Editor

PLOS ONE